# Optimal Flow Matching:
# Learning Straight Trajectories in Just One Step

**Nikita Kornilov**
Skolkovo Institute of Science and Technology
R.Center for AI, Innopolis University
Moscow Institute of Physics and Technology
`kornilov.nm@phystech.edu`

**Petr Mokrov**
Skolkovo Institute of Science and Technology
`petr.mokrov@skoltech.ru`

**Alexander Gasnikov**
Innopolis University
Moscow Institute of Physics and Technology
Steklov Mathematical Institute of RAS*
`gasnikov@yandex.ru`

**Alexander Korotin**
Skolkovo Institute of Science and Technology
Artificial Intelligence Research Institute
`a.korotin@skoltech.ru`

## Abstract

Over the several recent years, there has been a boom in development of Flow Matching (FM) methods for generative modeling. One intriguing property pursued by the community is the ability to learn flows with straight trajectories which realize the Optimal Transport (OT) displacements. Straightness is crucial for the fast integration (inference) of the learned flow's paths. Unfortunately, most existing flow straightening methods are based on non-trivial iterative FM procedures which accumulate the error during training or exploit heuristics based on minibatch OT. To address these issues, we develop and theoretically justify the novel **Optimal Flow Matching** (OFM) approach which allows recovering the straight OT displacement for the quadratic transport in just one FM step. The main idea of our approach is the employment of vector field for FM which are parameterized by convex functions. The code of our OFM implementation and the conducted experiments is available at https://github.com/Jhomanik/Optimal-Flow-Matching.

## 1 Introduction

Recent success in generative modeling [41, 17, 9] is mostly driven by Flow Matching (FM) [38] models. These models move a known distribution to a target one via ordinary differential equations (ODE) describing the mass movement. However, such processes usually have curved trajectories, resulting in time-consuming ODE integration for sampling. To overcome this issue, researches developed several improvements of the FM [39, 40, 48], which aim to recover more straight paths.

Rectified Flow (RF) method [39, 40] iteratively solves FM and gradually rectifies trajectories. Unfortunately, in each FM iteration, it **accumulates the error**, see [40, §2.2] and [39, §6]. This may spoil the performance of the method. The other popular branch of approaches to straighten trajectories is based on the connection between straight paths and Optimal Transport (OT) [60]. The main goal of OT is to find the way to move one probability distribution to another with the minimal effort. Such OT maps are usually described by ODEs with straight trajectories. In OT Conditional Flow Matching (OT-CFM) [48, 55], the authors propose to apply FM on top of OT solution between batches from considered distributions. Unfortunately, such a heuristic does not guarantee straight paths because of **minibatch OT biases**, see, e.g., [55, Figure 1, right] for the practical illustration.

---

*Russian Academy of Sciences

38th Conference on Neural Information Processing Systems (NeurIPS 2024).

**Contributions.** In this paper, we fix the above-mentioned problems of the straightening methods. We propose a novel Optimal Flow Matching (OFM) approach (§3) that after a **single** FM iteration obtains straight trajectories which can be simulated without ODE solving. It recovers OT flow for the quadratic transport cost function, i.e., it solves the Benamou–Brenier problem (Figure 1). We demonstrate the potential of OFM in the series of experiments and benchmarks (§4).

The main idea of our OFM is to consider during FM only specific vector fields which yield straight paths by design. These vector fields are the gradients of convex functions, which in practice are parametrized by Input Convex Neural Networks [3]. In OFM, one can optionally use minibatch OT or any other transport plan as the input, and this is completely theoretically justified.

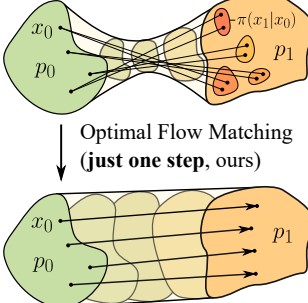

Optimal Flow Matching
(**just one step**, ours)

## 2 Background and Related Works

In this section, we provide all necessary backgrounds for the theory. First, we recall static (§2.1) and dynamic (§2.2) formulations of Optimal Transport and solvers (§2.3) for them. Then, we recall Flow Matching (§2.4.1) and flow straightening approaches: OT-CFM (§2.4.2) and Rectified Flow (§2.4.3).

**Notations.** For vectors $x, y \in \mathbb{R}^D$, we denote the inner product by $\langle x, y \rangle$ and the corresponding $\ell_2$ norm by $\|x\| := \sqrt{\langle x, x \rangle}$. We use $\mathcal{P}_{2,ac}(\mathbb{R}^D)$ to refer to the set of absolute continuous probability distributions with the finite second moment. For vector $x \in \mathbb{R}^D$ and distribution $p \in \mathcal{P}_{2,ac}(\mathbb{R}^D)$, notation $x \sim p$ means that $x$ is sampled from $p$. For the push-forward operator, we use symbol $\#$.

Figure 1: Our Optimal Flow Matching (OFM). For *any* initial transport plan $\pi$ between $p_0$ and $p_1$, OFM obtains *exactly straight* trajectories (in just a *single* FM loss minimization) which carry out the OT displacement for the quadratic cost function.

### 2.1 Static Optimal Transport

**Monge's and Kantorovich's formulations.** Consider two distributions $p_0, p_1 \in \mathcal{P}_{2,ac}(\mathbb{R}^D)$ and a cost function $c : \mathbb{R}^D \times \mathbb{R}^D \to \mathbb{R}$. Monge's Optimal Transport formulation is given by

$$\inf_{T\#p_0=p_1} \int_{\mathbb{R}^D} c(x_0, T(x_0)) p_0(x_0) dx_0, \tag{1}$$

where the infimum is taken over measurable functions $T : \mathbb{R}^D \to \mathbb{R}^D$ which satisfy the mass-preserving constraint $T\#p_0 = p_1$. Such functions are called transport maps. If there exists a transport map $T^*$ that achieves the infimum, then it is called the optimal transport map.

Since the optimal transport map $T^*$ in Monge's formulation may not exist, there is Kantorovich's relaxation for problem (1) which addresses this issue. Consider the set of transport plans $\Pi(p_0, p_1)$, i.e., the set of joint distributions on $\mathbb{R}^D \times \mathbb{R}^D$ which marginals are equal to $p_0$ and $p_1$, respectively. Kantorovich's Optimal Transport formulation is

$$\inf_{\pi \in \Pi(p_0,p_1)} \int_{\mathbb{R}^D \times \mathbb{R}^D} c(x_0, x_1) \pi(x_0, x_1) dx_0 dx_1. \tag{2}$$

With mild assumptions on $p_0, p_1$, the infimum is always achieved (possibly not uniquely). An optimal plan $\pi^* \in \Pi(p_0, p_1)$ is called an optimal transport plan. If optimal $\pi^*$ has the form $[\mathrm{id}, T^*]\#p_0$, then $T^*$ is the solution of Monge's formulation (1).

**Quadratic cost function.** In our paper, we mostly consider the quadratic cost function $c(x_0, x_1) = \frac{\|x_0 - x_1\|^2}{2}$. In this case, infimums in both Monge's and Kantorovich's OT are always uniquely attained [60, Brenier's Theorem 2.12]. They are related by $\pi^* = [\mathrm{id}, T^*]\#p_0$. Moreover, the optimal values of (1) and (2) are equal to each other. The square root of the optimal value is called the Wasserstein-2 distance $\mathbb{W}_2(p_0, p_1)$ between distributions $p_0$ and $p_1$, i.e.,

$$\mathbb{W}_2^2(p_0, p_1) := \min_{\pi \in \Pi(p_0,p_1)} \int_{\mathbb{R}^D \times \mathbb{R}^D} \frac{\|x_1 - x_0\|^2}{2} \pi(x_0, x_1) dx_0 dx_1$$

$$= \min_{T\#p_0=p_1} \int_{\mathbb{R}^D} \frac{\|x_0 - T(x_0)\|^2}{2} p_0(x_0) dx_0. \tag{3}$$

**Dual formulation.** For the quadratic cost, problem (3) has the equivalent dual form [60]:

$$\mathbb{W}_2^2(p_0, p_1) = \text{CONST}(p_0, p_1) - \min_{\text{convex } \Psi} \underbrace{\left[ \int_{\mathbb{R}^D} \Psi(x_0) p_0(x_0) dx_0 + \int_{\mathbb{R}^D} \overline{\Psi}(x_1) p_1(x_1) dx_1 \right]}_{=: \mathcal{L}_{OT}(\Psi)}, \quad (4)$$

where the minimum is taken over convex functions $\Psi(x) : \mathbb{R}^D \to \mathbb{R}$. Here $\overline{\Psi}(x_1) := \sup_{x_0 \in \mathbb{R}^D} [\langle x_0, x_1 \rangle - \Psi(x_0)]$ is the convex (Fenchel) conjugate function of $\Psi$. It is also convex.

The term $\text{CONST}(p_0, p_1)$ does not depend on $\Psi$. Therefore, the minimization (3) over transport plans $\pi$ is equivalent to the minimization of $\mathcal{L}_{OT}(\Psi)$ from (4) over convex functions $\Psi$. Moreover, the optimal transport map $T^*$ can be expressed via an optimal $\Psi^*$ (the *Brenier potential* [60]), namely,

$$T^* = \nabla \Psi^*. \quad (5)$$

## 2.2 Dynamic Optimal Transport

In [5], the authors show that the calculation of Optimal Transport map in (3) for the quadratic cost can be equivalently reformulated in a dynamic form. This form operates with a vector fields defining time-dependent mass transport instead of just static transport maps.

**Preliminaries.** We consider the fixed time interval $[0, 1]$. Let $u(t, \cdot) \equiv u_t(\cdot) : [0, 1] \times \mathbb{R}^D \to \mathbb{R}^D$ be a vector field and $\{\{z_t\}_{t \in [0,1]}\}$ be the set of random trajectories such that for each trajectory $\{z_t\}_{t \in [0,1]}$ the starting point $z_0$ is sampled from $p_0$ and $z_t$ satisfies the differential equation:

$$dz_t = u_t(z_t) dt, \quad z_0 \sim p_0. \quad (6)$$

In other words, the trajectory $\{z_t\}_{t \in [0,1]}$ is defined by its initial point $z_0 \sim p_0$ and goes along the speed vector $u_t(z_t)$. Under mild assumptions on $u$, for each initial $z_0$, the trajectory is unique.

Let $\phi^u(t, \cdot) \equiv \phi_t^u(\cdot) : [0, 1] \times \mathbb{R}^D \to \mathbb{R}^D$ denote the flow map, i.e., it is the function that maps the initial $z_0$ to its position at moment of time $t$ according to the ODE (6), i.e.,

$$d\phi_t^u(z_0) = u_t(\phi_t^u(z_0)), \quad \phi_0^u(z_0) = z_0. \quad (7)$$

If initial points $z_0$ of trajectories are distributed according to $p_0$, then (6) defines a distribution $p_t$ of $z_t$ at time $t$, which can be expressed via with the push-forward operator, i.e., $p_t^u := \phi_t^u \# p_0$.

**Benamou–Brenier problem.** Dynamic OT is the following minimization problem:

$$\mathbb{W}_2^2(p_0, p_1) = \inf_u \int_0^1 \int_{\mathbb{R}^D} \frac{\|u_t(x_t)\|_2^2}{2} \underbrace{\phi_t^u \# p_0(x_t)}_{:= p_t^u(x_t)} dx_t dt, \quad (8)$$

$$s.t. \quad \phi_1^u \# p_0 = p_1.$$

In (8), we look for the vector fields $u$ that define the flows which start at $p_0$ and end at $p_1$. Among such flows, we seek for the one which has the minimal kinetic energy over the entire time interval.

There is a connection between the static OT map $T^* = \nabla \Psi^*$ and the dynamic OT solution $u^*$. Namely, for every initial point $z_0$, the vector field $u^*$ defines a linear trajectory $\{z_t\}_{t \in [0,1]}$:

$$z_t = t \nabla \Psi^*(z_0) + (1 - t) z_0, \quad \forall t \in [0, 1]. \quad (9)$$

## 2.3 Continuous Optimal Transport Solvers

There exist a variety of continuous OT solvers [21, 52, 54, 43, 19, 14, 59, 15, 34, 51, 40, 36, 35, 13, 18, 57, 2, 55, 23, 44, 4, 20]. For a survey of solvers designed for OT with quadratic cost, see [33]. In this paper, we focus only on the most relevant ones, called the ICNN-based solvers [54, 33, 43, 2]. These solvers directly minimize objective $\mathcal{L}_{OT}$ from (4) parametrizing a class of convex functions with convex in input neural networks called ICNNs [3] (for more details, see "Parametrization of $\Psi$" in §3.2). Solvers details may differ, but the main idea remains the same. To calculate the conjugate function $\overline{\Psi}(x_1)$ at the point $x_1$, they solve the convex optimization problem from conjugate definition. Envelope Theorem [1] allows obtaining closed-form formula for the gradient of the loss.

## 2.4 Flow Matching Framework

In this section, we recall popular approaches [40, 39, 48] to find fields $u$ which transport a given probability distribution $p_0$ to a target $p_1$ and their relation to OT.

### 2.4.1 Flow Matching (FM)

To find such a field, one samples points $x_0, x_1$ from a transport plan $\pi \in \Pi(p_0, p_1)$, e.g., the independent plan $p_0 \times p_1$. The vector field $u$ is encouraged to follow the direction $x_1 - x_0$ of the linear interpolation $x_t = (1-t)x_0 + tx_1$ at any moment $t \in [0, 1]$, i.e., one solves:

$$\min_u \mathcal{L}_{FM}^\pi(u) := \int_0^1 \left\{ \int_{\mathbb{R}^D \times \mathbb{R}^D} \|u_t(x_t) - (x_1 - x_0)\|^2 \pi(x_0, x_1) dx_0 dx_1 \right\} dt, \, x_t = (1-t)x_0 + tx_1. \tag{10}$$

We denote the solution of (10) and the flow map (7) by $u^\pi$ and $\phi^\pi$, respectively. The concept of FM is depicted in Figure 2.

The intuition of this procedure is as follows: linear interpolation $x_t = (1-t)x_0 + tx_1$ is an intuitive way to move $p_0$ to $p_1$, but it requires knowing $x_1$. By fitting $u$ with the direction $x_1 - x_0$, one yields the vector field that can construct this interpolation without any information about $x_1$.

The set of trajectories $\{\{z_t\}_{t \in [0,1]}\}$ generated by $u_t^\pi$ (with $z_0 \sim p_0$) has a useful property: the flow map $\phi_1^\pi$ transforms distribution $p_0$ to distribution $p_1$ for any initial transport plan $\pi$. Moreover, marginal distribution $p_t = \phi_t^\pi \# p_0$ is equal to the distribution of linear interpolation $x_t = (1-t)x_0 + tx_1$ for any $t$ and $x_0, x_1 \sim \pi$. This feature is called the marginal preserving property.

To push point $x_0$ according to learned $u$, one needs to integrate ODE (6) via numerical solvers. The vector fields with straight (or nearly straight) paths incur much smaller time-discretization error and increase effectiveness of computations, which is in high demand for applications.

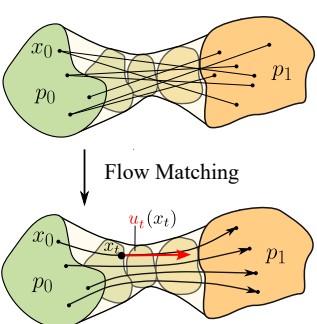

Figure 2: Flow Matching (FM) obtains a vector field $u$ moving $p_0$ to $p_1$. FM typically operates with the independent transport plan $\pi = p_0 \times p_1$.

Researchers noticed that some initial plans $\pi$ can result in more straight paths after FM rather than the standard independent plan $p_0 \times p_1$. The two most popular approaches to choose better plans are Optimal Transport Conditional Flow Matching [48, 55] and Rectified Flow [40].

### 2.4.2 Optimal Transport Conditional Flow Matching (OT-CFM)

If one uses the OT plan $\pi^*$ as the initial plan for FM, then it returns the Brenier's vector field $u^*$, which generates exactly straight trajectories (9). However, typically, the true OT plan $\pi^*$ is not available. In such a case, in order to achieve some level of straightness in the learned trajectories, a natural idea is to take the initial plan $\pi$ to be close to the optimal $\pi^*$. Inspired by this, the authors of OT-CFM [48, 55] take the advantage of minibatch OT plan approximation. Firstly, they independently sample batches of points from $p_0$ and $p_1$. Secondly, they join the batches together according to the discrete OT plan between them. The resulting joined batch is then used in FM. The concept of OT-CFM is depicted in Figure 3.

The main drawback of OT-CFM is that it recovers only biased dynamic OT solution. In order to converge to the true transport plan the batch size should be large [6], while with a growth of batch size computational time increases drastically [56]. In practice, batch sizes that ensure approximation good enough for applications are nearly infeasible to work with.

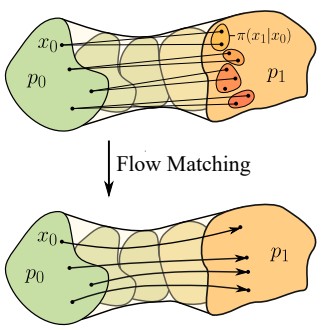

Figure 3: OT-CFM uses minibatch OT plan to obtain more straight trajectories.

### 2.4.3 Rectified Flow (RF)

In [40], the authors propose an iterative approach to refine the plan $\pi$, straightening the trajectories more and more with each iteration. Formally, Flow Matching procedure denoted by FM takes the transport plan $\pi$ as input and returns an optimal flow map via solving (10):

$$\phi^\pi := \text{FM}(\pi). \tag{11}$$

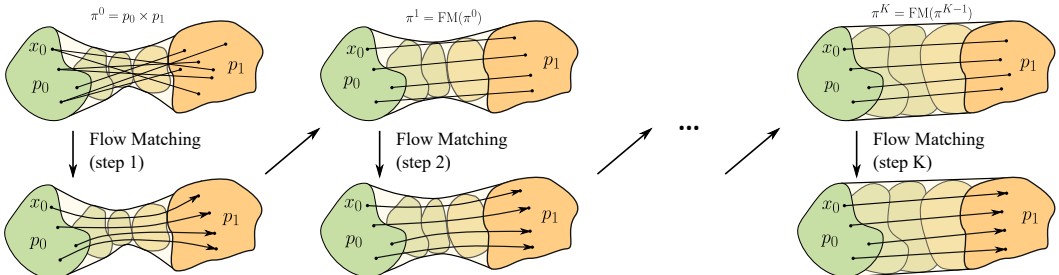

Figure 4: Rectified Flow iteratively applies FM to straighten the trajectories after each step.

One can iteratively apply FM to the initial transport plan (e.g., the independent plan), gradually rectifying it. Namely, Rectified Flow Algorithm on $K$-th iteration has the following update rule

$$\phi^{K+1} \quad = \quad \text{FM}(\pi^K), \quad \pi^{K+1} = [\text{id}, \phi^{K+1}] \# p_0, \tag{12}$$

where $\phi^K, \pi^K$ denote flow map and transport plan on $K$-th iteration, respectively.

With each new FM iteration, the generated trajectories $\{\{z_t\}_{t \in [0,1]}\}^K$ provably become more and more straight, i.e., error in approximation $z_t^K \approx (1-t)z_0^K + tz_1^K, \forall t \in [0,1]$ decreases as the number of iterations $K$ grows. The concept of RF is depicted on Figure 4.

The authors also notice that for any convex cost function $c$ the flow map $\phi_1^\pi$ from Flow Matching yields lower or equal transport cost than initial transport plan $\pi$:

$$\int_{\mathbb{R}^D} c(x_0, \phi_1^\pi(x_0))p_0(x_0)dx_0 \leq \int_{\mathbb{R}^D \times \mathbb{R}^D} c(x_0, x_1)\pi(x_0, x_1)dx_0 dx_1. \tag{13}$$

Intuitively, the transport costs are guaranteed to decrease because the trajectories of FM as solutions of well-defined ODE do not intersect each other, even if the initial lines connecting $x_0$ and $x_1$ can. With each iteration of RF (12), transport costs for all convex cost functions do not increase, but, for a given cost function, convergence to its own OT plan is not guaranteed. In [39], the authors address this issue and, for any particular convex cost function $c$, modify Rectified Flow to converge to OT map for $c$. In this modification, called $c$-Rectified Flow ($c$-RF), the authors slightly change the FM training objective and restrict the optimization domain only to potential vector fields $u_t(\cdot) = \nabla \bar{c}(\nabla f_t(\cdot))$, where $f_t(\cdot) : \mathbb{R}^D \to \mathbb{R}$ is an arbitrary time-dependent scalar valued function and $\bar{c}$ is the convex conjugate of the cost function $c$. In case of the quadratic cost function, the training objective remains the same, and the vector field $u_t$ is set as the simple gradient $\nabla f_t(\cdot)$ of the scalar valued function $f_t$.

Unfortunately, in practice, with each iteration ($c$-)RF accumulates error caused by inexactness from previous iterations, the issue mentioned in [39, §6, point 3]. Due to neural approximations, we can not get exact solution of FM (e.g., $\phi_1^K \# p_0 \neq p_1$), and this inexactness only grows with iterations. In addition, training of ($c$-)RF becomes non-simulation free after the first iteration, since to calculate the plan $\pi^{K+1} = [\text{id}, \phi^{K+1}] \# p_0$ it has to integrate ODE.

## 3 Optimal Flow Matching (OFM)

In this section, we provide the design of our novel Optimal Flow Matching algorithm (1) that fixes main problems of Rectified Flow and OT-CFM approaches described above. In theory, it obtains exactly **straight trajectories** and recovers the unbiased optimal transport map for the quadratic cost **just in one FM iteration** with **any** initial transport plan. Moreover, during inference, our OFM does not require solving ODE to transport points.

We discuss the theory behind our approach (§3.1), its practical implementation aspects (§3.2) and the relation to prior works (§3.3). All our proofs are located in Appendix A.

### 3.1 Theory: Deriving the Optimization Loss

We want to design a method of moving distribution $p_0$ to $p_1$ via exactly straight trajectories. Namely, we aim to obtain straight paths from the solution of the dynamic OT (8). Moreover, we want to limit ourselves to just one minimization iteration. Hence, we propose our novel Optimal Flow Matching (OFM) procedure satisfying the above-mentioned conditions. The main idea of our OFM is to

minimize the Flow Matching loss (10) not over all possible vector fields $u$, but only over specific *optimal* ones, which yield straight paths by construction and include the desired dynamic OT field $u^*$.

**Optimal vector fields.** We say that a vector field $u^\Psi$ is optimal if it generates linear trajectories $\{\{z_t\}_{t\in[0,1]}\}$ such that there exist a convex function $\Psi : \mathbb{R}^D \to \mathbb{R}$, which for any path $\{z_t\}_{t\in[0,1]}$ pushes the initial point $z_0$ to the final one as $z_1 = \nabla\Psi(z_0)$, i.e.,

$$z_t = (1-t)z_0 + t\nabla\Psi(z_0), \quad t \in [0,1].$$

The function $\Psi$ defines the ODE

$$dz_t = (\nabla\Psi(z_0) - z_0)dt, \quad z_t|_{t=0} = z_0. \tag{14}$$

Equation (14) does not provide a closed formula for $u^\Psi$ as it depends on $z_0$. The explicit formula is constructed as follows: for a time $t \in [0,1]$ and point $x_t$, we can find a trajectory $\{z_t\}_{t\in[0,1]}$ s.t.

$$x_t = z_t = (1-t)z_0 + t\nabla\Psi(z_0) \tag{15}$$

and recover the initial point $z_0$. We postpone the solution of this problem to §3.2. For now, we define the inverse of flow map (7) as $(\phi_t^\Psi)^{-1}(x_t) := z_0$ and the vector field $u_t^\Psi(x_t) := \nabla\Psi(z_0) - z_0 = \nabla\Psi((\phi_t^\Psi)^{-1}(x_t)) - (\phi_t^\Psi)^{-1}(x_t)$, which generates ODE (14), i.e., $dz_t = u_t^\Psi(z_t)dt$. The concept of optimal vector fields is depicted on Figure 5.

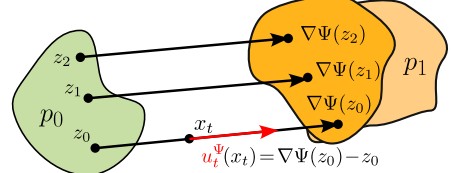

Figure 5: An Optimal Vector Field: a vector field $u^\Psi$ with straight paths is parametrized by a gradient of a convex function $\Psi$.

We highlight that the solution of dynamic OT lies in the class of optimal vector fields, since it generates linear trajectories (9) with the Brenier potential $\Psi^*$ (5).

**Training objective.** Our Optimal Flow Matching (OFM) approach is as follows: we restrict the optimization domain of FM (10) with fixed plan $\pi$ only to the optimal vector fields. We put the formula for the vector field $u_\Psi$ into FM loss from (10) and define our Optimal Flow Matching loss:

$$\mathcal{L}_{OFM}^\pi(\Psi) := \mathcal{L}_{FM}^\pi(u^\Psi) = \int_0^1 \left\{ \int_{\mathbb{R}^D \times \mathbb{R}^D} \|u_t^\Psi(x_t) - (x_1 - x_0)\|^2 \pi(x_0, x_1)dx_0 dx_1 \right\} dt, \tag{16}$$

$$x_t = (1-t)x_0 + tx_1.$$

Our Theorem 1 states that OFM solves the dynamic OT via single FM minimization for any initial $\pi$.

**Theorem 1** (OFM and OT connection). *Consider two distributions $p_0, p_1 \in \mathcal{P}_{ac,2}(\mathbb{R}^D)$ and **any** transport plan $\pi \in \Pi(p_0, p_1)$ between them. Then, the dual Optimal Transport loss $\mathcal{L}_{OT}$ (4) and Optimal Flow Matching loss $\mathcal{L}_{OFM}^\pi$ (16) have **the same minimizers**, i.e.,*

$$\underset{convex\ \Psi}{\arg\min}\ \mathcal{L}_{OFM}^\pi(\Psi) = \underset{convex\ \Psi}{\arg\min}\ \mathcal{L}_{OT}(\Psi).$$

### 3.2 Practical implementation aspects

In this subsection, we explain the details of optimization of our Optimal Flow Matching loss (16).

**Parametrization of $\Psi$.** In practice, we parametrize the class of convex functions with Input Convex Neural Networks (ICNNs) [3] $\Psi_\theta$ and parameters $\theta$. These are scalar-valued neural networks built in such a way that the network is convex in its input. They consist of fully-connected or convolution blocks, some weights of which are set to be non-negative in order to keep the convexity. In addition, activation functions are considered to be only non-decreasing and convex in each input coordinate. These networks are able to support most of the popular training techniques (e.g., gradient descent optimization, dropout, skip connection, etc.). In Appendix B, we discuss the used architectures.

**OFM loss calculation.** We provide an explicit formula for gradient of OFM loss (16).

**Proposition 1** (Explicit Loss Gradient Formula). *The gradient of $\mathcal{L}_{OFM}^\pi$ can be calculated as*

$$z_0 = \text{NO-GRAD}\left\{ (\phi_t^{\Psi_\theta})^{-1}(x_t) \right\},$$

$$\frac{d\mathcal{L}_{OFM}^\pi}{d\theta} := \frac{d}{d\theta}\mathbb{E}_{t;x_0,x_1\sim\pi}\left\langle \text{NO-GRAD}\left\{ 2\left(t\nabla^2\Psi_\theta(z_0) + (1-t)I\right)^{-1}\frac{(x_0 - z_0)}{t} \right\}, \nabla\Psi_\theta(z_0) \right\rangle,$$

*where variables under* NO-GRAD *remain constants during differentiation.*

**Flow map inversion.** In order to find the initial point $z_0 = (\phi_t^\Psi)^{-1}(x_t)$, we note that (15)

$$x_t = (1 - t)z_0 + t\nabla\Psi(z_0)$$

is equivalent to

$$\nabla\left(\frac{(1-t)}{2}\|\cdot\|^2 + t\Psi(\cdot) - \langle x_t, \cdot\rangle\right)(z_0) = 0.$$

The function under gradient operator $\nabla$ has minimum at the required point $z_0$, since at $z_0$ the gradient of it equals 0. If $t < 1$ the function is at least $(1-t)$-strongly convex, and the minimum is unique. The case $t = 1$ is negligible in practice, since it has zero probability to appear during training.

We can reduce the problem of inversion to the following minimization subproblem

$$(\phi_t^\Psi)^{-1}(x_t) = \arg\min_{z_0 \in \mathbb{R}^D}\left[\frac{(1-t)}{2}\|z_0\|^2 + t\Psi(z_0) - \langle x_t, z_0\rangle\right]. \tag{17}$$

Optimization subproblem (17) is at least $(1-t)$**-strongly convex** and can be effectively solved for any given point $x_t$ (in comparison with typical non-convex optimization tasks).

**Algorithm.** The Optimal Flow Matching pseudocode is presented in listing 1. We estimate math expectation over plan $\pi$ and time $t$ with uniform distribution on $[0, 1]$ via unbiased Monte Carlo.

---

**Algorithm 1** Optimal Flow Matching

---

**Input:** Initial transport plan $\pi \in \Pi(p_0, p_1)$, number of iterations $K$, batch size $B$, optimizer $Opt$, sub-problem optimizer $SubOpt$, ICNN $\Psi_\theta$

1: **for** $k = 0, \ldots, K - 1$ **do**
2:     Sample batch $\{(x_0^i, x_1^i)\}_{i=1}^B$ of size $B$ from plan $\pi$;
3:     Sample times batch $\{t^i\}_{i=1}^B$ of size $B$ from $U[0, 1]$;
4:     Calculate linear interpolation $x_{t^i}^i = (1 - t^i)x_0^i + t^i x_1^i$ for all $i \in \overline{1, B}$;
5:     Find the initial points $z_0^i$ via solving the convex problem with $SubOpt$:

$$z_0^i = \text{NO-GRAD}\left\{\arg\min_{z_0^i}\left[\frac{(1-t^i)}{2}\|z_0^i\|^2 + t^i\Psi_\theta(z_0^i) - \langle x_{t^i}^i, z_0^i\rangle\right]\right\};$$

6:     Calculate loss $\hat{\mathcal{L}}_{OFM}$

$$\hat{\mathcal{L}}_{OFM} = \frac{1}{B}\sum_{i=1}^B \left\langle \text{NO-GRAD}\left\{2\left(t^i\nabla^2\Psi_\theta(z_0^i) + (1-t^i)I\right)^{-1}\frac{(x_0^i - z_0^i)}{t^i}\right\}, \nabla\Psi_\theta(z_0^i)\right\rangle;$$

7:     Update parameters $\theta$ via optimizer $Opt$ step with $\frac{d\hat{\mathcal{L}}_{OFM}}{d\theta}$;
8: **end for**

---

### 3.3 Relation to Prior Works

In this subsection, we compare our Optimal Flow Matching and previous straightening approaches. One unique feature of OFM is that it works only with flows which have straight paths by design and does not require ODE integration to transport points. Other methods may result in non-straight paths during training, and they still have to solve ODE even with near-straight paths.

**OT Solvers** [54, 43, 2]. According to Theorem 1, our OFM and dual OT solvers basically minimize the same OT loss (4). However, our OFM actively utilizes the temporal component of the dynamic process. It allows us to pave a novel theoretical bridge between OT and FM. Such a direct connection can lead to the adoption of the strengths of both methods and a deeper understanding of them.

**OT-CFM** [48, 55]. Unlike our OFM approach, OT-CFM method retrieves biased OT solution, and the recovery of straight paths is not guaranteed. In OT-CFM, minibatch OT plan appears as a heuristic that helps to get better trajectories in practice. In contrast, usage of **any** initial transport plan $\pi$ in our OFM is completely justified in Theorem 1.

**Rectified Flow** [40, 39]. In Rectified Flows [40], the authors iteratively apply Flow Matching to refine the obtained trajectories. However, in each iteration, RF accumulates error since one may not

learn the exact flow due to neural approximations. In addition, RF does not guarantee convergence to the OT plan for the quadratic cost. The $c$-Rectified Flow [39] modification can converge to the OT plan for any cost function $c$, but still remains iterative. In addition, RF and $c$-RF both requires ODE simulation after the first iteration to continue training. In OFM, we work only with the quadratic cost function, but retrieve its OT solution in **just one FM iteration** without simulation of the trajectories.

**Light and Optimal Schrödinger Bridge.** In [22], the authors observe the relation between Entropic Optimal Transport (EOT) [42, 12] and Bridge Matching (BM) [53] problems. These are stochastic analogs of OT and FM, respectively. In EOT and BM, instead of deterministic ODE and flows, one considers stochastic processes with non-zero stochasticity. The authors prove that, during BM, one can restrict considered processes only to the specific ones and retrieve the solution of EOT. Hypothetically, our OT/FM case is a limit of their EOT/BM case when the stochasticity tends to zero. Proofs in [22] for EOT are based on sophisticated KL divergence properties. We do not know whether our results for OFM can be derived by taking the limit of their stochastic case. To derive the properties of our OFM, we use **completely different proof techniques** based on computing integrals over curves rather than KL-based techniques. Besides, in practice, the authors of [22] mostly focus on Gaussian mixture parametrization while our method allows using neural networks (ICNNs).

### 3.4 Theory: properties of OFM

In this subsection, we provide the OFM's theoretical properties, which give an intuition for understanding of its main working principles and behavior.

**Proposition 2** (Simplified OFM Loss). *We can simplify* (16) *to a more suitable form:*

$$\mathcal{L}_{OFM}^{\pi}(\Psi) = \int_0^1 \left\{ \int_{\mathbb{R}^D \times \mathbb{R}^D} \left\| \frac{(\phi_t^{\Psi})^{-1}(x_t) - x_0}{t} \right\|^2 \pi(x_0, x_1) dx_0 dx_1 \right\} dt, \ x_t = (1-t)x_0 + tx_1. \quad (18)$$

The simplified form (18) shows that OFM loss actually measures how well $\Psi$ restores initial points $x_0$ of linear interpolations depending on future point $x_t$ and time $t$.

**Generative properties of OFM.** In this paragraph, we provide another view on our OFM approach. In our OFM, we aim to construct a vector field $u$ which is as close to the dynamic OT field $u^*$ as possible. We can use the least square regression to measure the distance between them:

$$\mathrm{DIST}(u, u^*) := \int_0^1 \int_{\mathbb{R}^D} \|u_t(x_t) - u_t^*(x_t)\|^2 \underbrace{\phi_t^* \# p_0(x_t)}_{:= p_t^*(x_t)} dx_t dt. \quad (19)$$

**Proposition 3** (Intractable Distance). *The distance* $\mathrm{DIST}(u, u^*)$ *between an arbitrary vector field* $u$ *and OT field* $u^*$ *equals to the FM loss from* (10) *with the optimal plan* $\pi^*$, *i.e.,*

$$\mathrm{DIST}(u, u^*) = \mathcal{L}_{FM}^{\pi^*}(u) - \underbrace{\mathcal{L}_{FM}^{\pi^*}(u^*)}_{=0}.$$

We can not minimize intractable $\mathrm{DIST}(u, u^*)$ since the optimal plan $\pi^*$ is unknown. In OT-CFM [55], authors heuristically approximate $\pi^*$ in $\mathcal{L}_{FM}^{\pi^*}(u)$, but obtain biased solution. Surprisingly, for the *optimal* vector fields, the distance can be calculated explicitly via **any** known plan $\pi$.

**Proposition 4** (Tractable Distance For OFM). *The distance* $\mathrm{DIST}(u^{\Psi}, u^{\Psi^*})$ *between an **optimal** vector field* $u^{\Psi}$ *generated by a convex function* $\Psi$ *and the vector field* $u^{\Psi^*}$ *with the Brenier potential* $\Psi^*$ *can be evaluated directly via OFM loss* (16) *and **any** plan* $\pi$:

$$\mathrm{DIST}(u^{\Psi}, u^{\Psi^*}) = \mathcal{L}_{FM}^{\pi}(u^{\Psi}) - \mathcal{L}_{FM}^{\pi}(u^{\Psi^*}) = \mathcal{L}_{OFM}^{\pi}(\Psi) - \mathcal{L}_{OFM}^{\pi}(\Psi^*). \quad (20)$$

In (20), the first term is our tractable OFM loss, and the second term does not depend on $\Psi$. Hence, during the whole minimization process in our OFM, we gradually lower the distance (19) between the current vector field and the dynamic OT field up to the complete match.

## 4 Experimental Illustrations

In this section, we showcase the performance of our Optimal Flow Matching method on illustrative 2D scenario (§4.1) and Wasserstein-2 benchmark [33] (§4.2). Finally, we apply our approach for solving high-dimensional unpaired image-to-image translation in the latent space of pretrained ALAE autoencoder (§4.3). The `PyTorch` implementation of our method is publicly available at

The technical details of our experiments (architectures, hyperparameters) are in the Appendix B.

## 4.1 Illustrative 2D Example

In this subsection, we illustrate the proof-of-concept of our OFM on 2D setup and demonstrate that OFM's solutions do not depend on the initial transport plan $\pi$. We run our Algorithm 1 between a standard Gaussian $p_0 = \mathcal{N}(0, I)$ and a Mixture of eight Gaussians $p_1$ depicted in the Figure 6a. We consider different stochastic plans $\pi$: independent plan $p_0 \times p_1$ (Figure 6b), minibatch and *anti*minibatch (Figures 6c, 6d) discrete OT (quadratic cost) with batch size $B_{\text{mb}} = 64$. In the **anti**minibatch case, we compose the pairs of source and target points by solving discrete OT with **minus** quadratic cost $-\|x - y\|_2^2$. The fitted OFM maps and trajectories are presented in Figure 6. We empirically see that our OFM finds the *same solution for all initial plans $\pi$*.

For completeness, in Appendix B.2, we apply these plans to the original FM (10), and show that, in comparison with our OFM, the resulting paths obtained by FM considerably depend on the plan.

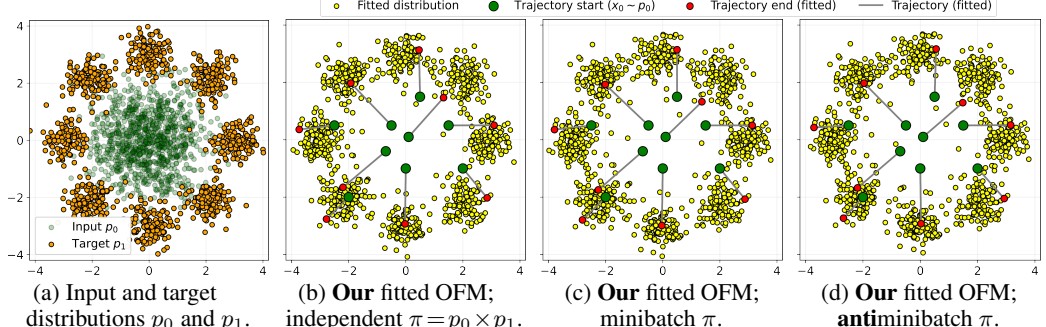

(a) Input and target distributions $p_0$ and $p_1$.  (b) **Our** fitted OFM; independent $\pi = p_0 \times p_1$.  (c) **Our** fitted OFM; minibatch $\pi$.  (d) **Our** fitted OFM; **anti**minibatch $\pi$.

Figure 6: Performance of **our** Optimal Flow Matching on *Gaussian→Eight Gaussians* 2D setup.

## 4.2 High-dimensional OT Benchmarks

In this subsection, we quantitatively compare our OFM and other methods testing their ability to solve OT. We run our OFM, FM based methods and OT solvers on OT Benchmark [33]. The authors provide high-dimensional continuous distributions $p_0, p_1$ for which the ground truth OT map $T^*$ for the quadratic cost is known by the construction. To assess the quality of retrieved transport maps, we use standard *unexplained variance percentage* $\mathcal{L}^2\text{-UVP}(T) := 100 \cdot \|T - T^*\|_{\mathcal{L}^2(p_0)}^2 / \text{Var}(p_1)\%$ [33]. It directly computes the normalized squared error between OT map $T^*$ and learned map $T$.

**Competitors.** We evaluate Conditional Flow Matching (OT-CFM), Rectified Flow (RF), $c$-Rectified Flow ($c$-RF), the most relevant OT solver MMv-1 [54] and its amortized version from [2]. In [54] and [2], the authors directly minimize the dual formulation loss $\mathcal{L}_{OT}$ (4) by parametrizing $\Psi$ with ICNNs and calculating $\overline{\Psi}(x_1)$ via solving a convex optimization subproblem. The latter is similar to our inversion (17). Additionally, in [2], the authors use MLPs to parametrize $\Psi$, and we include these results as well. Following [33], we also provide results for a linear OT map (baseline) which translates means and variances of distributions to each other. For our OFM, we consider two initial plans: independent plan (Ind) and minibatch OT (MB), the batch size for the latter is $B_{\text{mb}} = 64$.

The overall results are presented in Table 1. More details are given in Appendix B.3.

| Solver | Solver type | $D=2$ | $D=4$ | $D=8$ | $D=16$ | $D=32$ | $D=64$ | $D=128$ | $D=256$ |
|---|---|---|---|---|---|---|---|---|---|
| MMv1$^*$ [54] | | 0.2 | 1.0 | 1.8 | 1.4 | 6.9 | 8.1 | 2.2 | 2.6 |
| Amortization, ICNN$^{**}$ [2] | Dual OT solver | 0.26 | 0.78 | 1.6 | 1.1 | 1.9 | 4.2 | 1.6 | 2.0 |
| Amortization, MLP$^{**}$ [2] | | 0.03 | 0.22 | 0.6 | 0.8 | 2.0 | 2.1 | 0.67 | 0.59 |
| Linear$^*$ [33] | Baseline | 14.1 | 14.9 | 27.3 | 41.6 | 55.3 | 63.9 | 63.6 | 67.4 |
| OT-CFM [55] | | 0.16 | 0.73 | 2.27 | 4.33 | 7.9 | 11.4 | 12.1 | 27.5 |
| RF [40] | | 8.58 | 49.46 | 51.25 | 63.33 | 63.52 | 85.13 | 84.49 | 83.13 |
| $c$-RF [39] | Flow Matching | 1.56 | 13.11 | 17.87 | 35.39 | 48.46 | 66.52 | 68.08 | 76.48 |
| OFM Ind **(Ours)** | | 0.19 | 0.61 | 1.4 | 1.1 | 1.47 | 8.35 | 1.96 | 3.96 |
| OFM MB **(Ours)** | | **0.15** | **0.52** | **1.2** | **1.0** | **1.2** | **7.2** | **1.5** | **2.9** |

Table 1: $\mathcal{L}^2-$UVP values of solvers fitted on high-dimensional benchmarks in dimensions $D = 2, 4, 8, 16, 32, 64, 128, 256$. The best metric over *Flow Matching based* methods is **bolded**. $^*$ Metrics are taken from [33]. $^{**}$ Metrics are taken from [2].

**Results.** Among FM-based methods, our OFM with any plan demonstrates the best results. For all plans, OFM convergences to close final solutions and metrics. Minibatch plan provides a little bit better results, especially in high dimensions. In theory, the OFM results for any plan $\pi$ must be similar. However, in stochastic optimization, plans with large variance yield convergence to slightly worse solutions.

MLP-based OT solver usually beats our OFM, since MLPs do not have ICNNs' limitations in practice. However, usage of MLP is an empirical trick and is not completely justified. We also run OFM with MLP instead ICNN, and, unfortunately, the method fails to converge.

RF demonstrates worse performance than even linear baseline, but it is ok since it is not designed to solve $\mathbb{W}_2$ OT. In turn, $c$-RF works better, but rapidly deteriorates with increasing dimensions. OT-CFM demonstrates the best results among baseline FM-based methods, but still underperforms compared to our OFM solver in high dimensions.

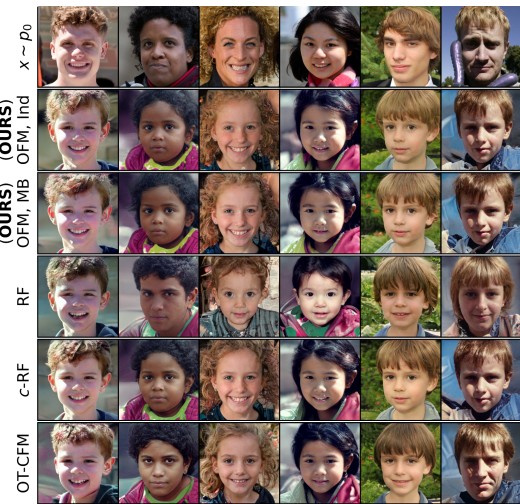

Figure 7: Unpaired I2I *Adult→Child* by FM solvers, ALAE $1024 \times 1024$ FFHQ latent space.

### 4.3 Unpaired Image-to-image Transfer

Another task that involves learning a translation between two distributions is unpaired image-to-image translation [63]. We follow the setup of [32] where translation is computed in the 512 dimensional latent space of the pre-trained ALAE autoencoder [46] on $1024 \times 1024$ FFHQ dataset [29]. In particular, we split the train FFHQ sample (60K faces) into *children* and *adults* subsets and consider the corresponding ALAE latent codes as the source and target distributions $p_0$ and $p_1$. At the inference stage, we take a new (unseen) *adult* face from a test FFHQ sample, extract its latent code, process with our learned model and then decode back to the image space. The qualitative translation results and FID metric [25] are presented in Figure 7 and Table 2, respectively.

| Method | OFM, Ind (Ours) | OFM, MB (Ours) | RF | $c-$RF | OT-CFM |
|---|---|---|---|---|---|
| FID | 11.8 | **11.0** | 21.0 | 13.5 | 12.9 |

Table 2: FID metric on Adult→Child translation task for the Flow Matching based methods.

The batch size for minibatch OT methods ($\lfloor$OFM, MB$\rceil$, $\lfloor$OT-CFM$\rceil$) is $B_{\text{mb}} = 128$. Our OFM converges to nearly the same solution for both independent and MB plans, and demonstrates qualitatively plausible translations. The most similar results to our method are demonstrated by $\lfloor c$-RF$\rceil$. Similar to OFM, this method (in the limit of RF steps) also recovers the quadratic OT mapping.

## 5 Discussion

**Potential impact.** We believe that our novel theoretical results have a huge potential for improving modern flow matching-based methods and inspiring the community for further studies. We think this is of high importance especially taking into account that modern generative models start to extensively use flow matching methods [61, 41, 17].

**Limitations and broader impact** are discussed in Appendix C.

## 6 Acknowledgement

The work of N. Kornilov has been financially supported by The Analytical Center for the Government of the Russian Federation (Agreement No. 70-2021-00143 01.11.2021, IGK 000000D730324P540002).

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

# A Proofs and auxiliary statements

In this section, we place the proofs of all our results from the main manuscript and some auxiliary results. We reorder the proofs of Prop. 1 and Prop. 2 since the former is based on the latter.

Note that in all of our theoretical derivations, if not stated explicitly, we assume the differentiability of convex potential $\Psi$ at given points $z_0, x_t$ etc. This assumption is done for simplicity and does not spoil our theory. The convex functions are known to be differentiable almost surely w.r.t Lebesgue measure [50]. Therefore, since we consider absolutely continuous reference distributions $p_0, p_1$ (see §2.1), the differentiability of $\Psi$ at the considered points also holds almost surely.

**Proof of Proposition 2** (Simplified OFM Loss)

*Proof.* By definition $\mathcal{L}_{OFM}^{\pi}(\Psi)$ equals to

$$\mathcal{L}_{OFM}^{\pi}(\Psi) := \int_0^1 \left\{ \int_{\mathbb{R}^D \times \mathbb{R}^D} \|u_t^{\Psi}(x_t) - (x_1 - x_0)\|^2 \pi(x_0, x_1) dx_0 dx_1 \right\} dt, x_t = (1-t)x_0 + tx_1. \quad (21)$$

For fixed points $x_0, x_1$ and time $t$ in integrand, we find a point $z_0 = (\phi_t^{\Psi})^{-1}(x_t)$ such that in moment $t \in [0, 1]$ it is transported to point $x_t = (1-t)x_0 + tx_1$. This point $z_0$ satisfies equality

$$x_t = t\nabla\Psi(z_0) + (1-t)z_0.$$

We define the vector field $u_t^{\Psi}$ as

$$u_t^{\Psi}(x_t) = \nabla\Psi(z_0) - z_0 = \frac{x_t - z_0}{t}.$$

Putting $u_t^{\Psi}(x_t)$ in the integrand of (21), we obtain simplified integrand

$$\|x_1 - x_0 - u_t^{\Psi}(x_t)\|^2 = \left\| x_1 - x_0 - \left(\frac{x_t - z_0}{t}\right) \right\|^2$$

$$= \frac{1}{t^2}\|tx_1 - tx_0 - ((1-t)x_0 + tx_1) + z_0\|^2$$

$$= \frac{1}{t^2}\|z_0 - x_0\|^2 = \left\| \frac{(\phi_t^{\Psi})^{-1}(x_t) - x_0}{t} \right\|^2.$$

$\square$

**Proof of Proposition 1** (Explicit Loss Gradient Formula)

*Proof.* Point $z_0 = (\phi_t^{\Psi_\theta})^{-1}(x_t)$ now depends on parameters $\theta$. We differentiate the integrand from the simplified OFM loss (18) for fixed points $x_0, x_1$ and time $t$, i.e.,

$$d\left(\frac{1}{t^2}\|z_0 - x_0\|^2\right) = 2\left\langle \frac{z_0 - x_0}{t^2}, \frac{dz_0}{d\theta}d\theta \right\rangle. \quad (22)$$

For point $z_0$, the equation (27) holds true:

$$x_t = (1-t)z_0 + t\nabla\Psi_\theta(z_0). \quad (23)$$

We differentiate (23) w.r.t. $\theta$ and obtain

$$0 = (1-t)\frac{dz_0}{d\theta} + t\nabla^2\Psi_\theta(z_0)\frac{dz_0}{d\theta} + t\frac{\partial\nabla\Psi_\theta}{\partial\theta}(z_0) \Rightarrow$$

$$\frac{dz_0}{d\theta} = -\left(t\nabla^2\Psi_\theta(z_0) + (1-t)I\right)^{-1} \cdot t\frac{\partial\nabla\Psi_\theta}{\partial\theta}(z_0).$$

Therefore, we have

$$(22) = \left\langle 2\frac{x_0 - z_0}{t}, \left(t\nabla^2\Psi_\theta(z_0) + (1-t)I\right)^{-1}\frac{\partial\nabla\Psi_\theta}{\partial\theta}(z_0)d\theta \right\rangle$$

$$= \left\langle 2\left(t\nabla^2\Psi_\theta(z_0) + (1-t)I\right)^{-1}\frac{(x_0 - z_0)}{t}, \frac{\partial\nabla\Psi_\theta}{\partial\theta}(z_0)d\theta \right\rangle. \quad (24)$$

Now the differentiation over $\theta$ is located only in the right part of (24) in the term $\frac{\partial \nabla \Psi_\theta}{\partial \theta}$. Hence, point $z_0$ and the left part of (24) can be considered as constants during differentiation. To get the gradient of OFM loss we also need to take math expectation over plan $\pi$ and time $t$. $\qquad \square$

The following two Lemmas are used to prove our main theoretical result, Theorem 1.

**Lemma 1** (Properties of convex functions and their conjugates). *Let $\Psi : \mathbb{R}^D \to \mathbb{R}$ be a convex function; $x_0, x_1 \in \mathbb{R}^D$. Let $\Psi$ and $\overline{\Psi}$ be differentiable at $x_0$ and $x_1$ correspondingly. Then the following statements are equivalent:*

*(i)* $x_1 = \nabla \Psi(x_0)$;

*(ii)* $x_0 = \arg\max_{z \in \mathbb{R}^D} \{\langle x_1, z \rangle - \Psi(z)\}$;

*(iii)* Fenchel-Young's equality: $\Psi(x_0) + \overline{\Psi}(x_1) = \langle x_1, x_0 \rangle$;

*(iv)* $x_0 = \nabla \overline{\Psi}(x_1)$;

*(v)* $x_1 = \arg\max_{z \in \mathbb{R}^D} \{\langle z, x_0 \rangle - \overline{\Psi}(z)\}$;

*Proof.* The lemma is a simplified version of [47, Theorem 1.16.4]. Also, the proof can be constructed by combining facts from [7, §3.3]. $\qquad \square$

**Lemma 2** (Main Integration Lemma). *For any two points $x_0, x_1 \in \mathbb{R}^D$ and a convex function $\Psi$, the following equality holds true:*

$$\int_0^1 \|u_t^\Psi(x_t) - (x_1 - x_0)\|^2 dt = 2 \cdot [\Psi(x_0) + \overline{\Psi}(x_1) - \langle x_0, x_1 \rangle], \qquad (25)$$

*where $x_t = tx_0 + (1 - t)x_1$.*

*Proof.* Following Proposition 2, we use the simplified loss form, i.e.,

$$\|u_t^\Psi(x_t) - (x_1 - x_0)\|^2 = \frac{1}{t^2}\|z_0 - x_0\|^2, \qquad (26)$$

where $z_0 = z_0(t) = (\phi_t^\Psi)^{-1}(x_t)$ satisfies the equality:

$$x_t = t\nabla\Psi(z_0) + (1 - t)z_0. \qquad (27)$$

Next, we substitute (26) into rhs of (25) integrate w.r.t. time $t$ from 0 excluding to 1 excluding (This exclusion does not change the integral):

$$\int_0^1 \|u_t^\Psi(x_t) - (x_1 - x_0)\|^2 dt = \int_0^1 \frac{1}{t^2}\|z_0 - x_0\|^2 dt. \qquad (28)$$

To further simplify (28) we need some preliminary work. Following (27) we note:

$$
\begin{aligned}
x_t = t\nabla\Psi(z_0) + (1 - t)z_0 &= (1 - t)x_0 + tx_1 \Rightarrow \\
t(\nabla\Psi(z_0) - x_1) &= (1 - t)(x_0 - z_0) \Rightarrow \\
(\nabla\Psi(z_0) - x_1) &= \left(\frac{1 - t}{t}\right)(x_0 - z_0) \Rightarrow \qquad (29) \\
\|\nabla\Psi(z_0) - x_1\|^2 &= \frac{(1 - t)^2}{t^2}\|z_0 - x_0\|^2. \qquad (30)
\end{aligned}
$$

Changing in (28) time variable $t$ to $s = \frac{t}{1-t}$, $ds = \frac{dt}{(1-t)^2}$ and substitution of (30) yield:

$$\int_0^1 \frac{1}{t^2}\|z_0(t) - x_0\|^2 dt = \int_0^1 \frac{(1 - t)^2}{t^2}\|z_0(t) - x_0\|^2 \frac{dt}{(1-t)^2} = \int_0^\infty \|\nabla\Psi(z_0(s)) - x_1\|^2 ds. \quad (31)$$

We notice that set of points $z_0(s(t)) = (\phi_t^\Psi)^{-1}(x_t), t \in (0,1)$ forms a curve in $\mathbb{R}^D$ with parameter $t$ (or $s(t)$). Now we are to process formula (31) by switching from the integration w.r.t. parameter $s$ to the integration along this curve. To do it properly we need two things:

1. Limits of integration. The limits of integration along the curve $z_0(t)$ are:

$$z_0(t)|_{t=0} = x_0,$$
$$z_0(t)|_{t=1} = (\nabla\Psi)^{-1}(x_1) \overset{\text{Lemma 1; (i), (iv)}}{=} \nabla\overline{\Psi}(x_1). \tag{32}$$

2. Expression under integral sign w.r.t. differential $dz_0$. Starting with (29), we derive:

$$
\begin{aligned}
(29) \quad \Rightarrow \quad s(\nabla\Psi(z_0) - x_1) &= (x_0 - z_0) \Rightarrow \\
d[s(\nabla\Psi(z_0) - x_1)] &= d[x_0 - z_0] \Rightarrow \\
s\nabla^2\Psi(z_0)dz_0 + (\nabla\Psi(z_0) - x_1)ds &= -dz_0 \Rightarrow \\
(\nabla\Psi(z_0) - x_1)ds &= -(s\nabla^2\Psi(z_0) + I)dz_0.
\end{aligned}
\tag{33}
$$

Now we proceed with (31):

$$
\begin{aligned}
(31) &= \int_0^\infty \langle \nabla\Psi(z_0) - x_1, \nabla\Psi(z_0) - x_1 \rangle \, ds \\
&\overset{(33)}{=} \int_{z_0} \langle x_1 - \nabla\Psi(z_0), (s\nabla^2\Psi(z_0) + I)dz_0 \rangle \\
&= \int_{z_0} \langle x_1 - \nabla\Psi(z_0), dz_0 \rangle + \int_{z_0} \langle s(x_1 - \nabla\Psi(z_0)), \nabla^2\Psi(z_0)dz_0 \rangle \\
&\overset{(29)}{=} \int_{z_0} \langle x_1 - \nabla\Psi(z_0), dz_0 \rangle + \int_{z_0} \langle z_0 - x_0, \nabla^2\Psi(z_0)dz_0 \rangle.
\end{aligned}
\tag{34}
$$

We notice that

$$
\begin{aligned}
d\langle z_0, \nabla\Psi(z_0) \rangle &= \langle z_0, \nabla^2\Psi(z_0)dz_0 \rangle + \langle dz_0, \nabla\Psi(z_0) \rangle \Rightarrow \\
\langle z_0, \nabla^2\Psi(z_0)dz_0 \rangle &= d\langle z_0, \nabla\Psi(z_0) \rangle - \langle \nabla\Psi(z_0), dz_0 \rangle.
\end{aligned}
$$

As a consequence, we further proceed with (34):

$$
\begin{aligned}
(34) &= \int_{z_0} \langle x_1 - \nabla\Psi(z_0), dz_0 \rangle + \int_{z_0} \langle z_0 - x_0, \nabla^2\Psi(z_0)dz_0 \rangle \\
&= \int_{z_0} \langle x_1, dz_0 \rangle - \int_{z_0} \langle \nabla\Psi(z_0), dz_0 \rangle \\
&\quad + \int_{z_0} d\langle z_0, \nabla\Psi(z_0) \rangle - \int_{z_0} \langle \nabla\Psi(z_0), dz_0 \rangle - \int_{z_0} \langle x_0, \nabla^2\Psi(z_0)dz_0 \rangle \\
&= \int_{z_0} \langle x_1, dz_0 \rangle - 2\int_{z_0} \langle \nabla\Psi(z_0), dz_0 \rangle + \int_{z_0} d\langle z_0, \nabla\Psi(z_0) \rangle - \int_{z_0} \langle x_0, \nabla^2\Psi(z_0)dz_0 \rangle.
\end{aligned}
\tag{35}
$$

Under all integrals we have closed form differentials

$$
\begin{aligned}
\langle x_1, dz_0 \rangle &= d\langle x_1, z_0 \rangle, \\
\langle \nabla\Psi(z_0), dz_0 \rangle &= d\Psi(z_0), \\
\langle x_0, \nabla^2\Psi(z_0)dz_0 \rangle &= d\langle x_0, \nabla\Psi(z_0) \rangle.
\end{aligned}
$$

We integrate them from initial point $x_0$ to the final $\nabla\overline{\Psi}(x_1)$ according to limits (32) and get

$$
\begin{aligned}
(35) &= \int_{z_0} d\langle x_1, z_0 \rangle - 2\int_{z_0} d\Psi(z_0) + \int_{z_0} d\langle z_0, \nabla\Psi(z_0) \rangle - \int_{z_0} d\langle x_0, \nabla\Psi(z_0) \rangle \\
&= \langle x_1, \nabla\overline{\Psi}(x_1) \rangle - \langle x_1, x_0 \rangle + 2(\Psi(x_0) - \Psi(\nabla\overline{\Psi}(x_1))) + \langle (\nabla\overline{\Psi}(x_1), \nabla\Psi(\nabla\overline{\Psi}(x_1)) \rangle \\
&\quad - \langle x_0, \nabla\Psi(x_0) \rangle + \langle x_0, \nabla\Psi(x_0) \rangle - \langle x_0, \nabla\Psi(\nabla\overline{\Psi}(x_1)) \rangle.
\end{aligned}
\tag{36}
$$

Now we use properties of conjugate functions (Lemma (1)):

$$\Psi(\nabla\overline{\Psi}(x_1)) \overset{\text{(iv) + (iii)}}{=} \langle\nabla\overline{\Psi}(x_1), x_1\rangle - \overline{\Psi}(x_1),$$

$$\nabla\Psi(\nabla\overline{\Psi}(x_1)) \overset{\text{(iv) + (i)}}{=} x_1.$$

This allows us to simplify (36):

$$
\begin{aligned}
(36) &= \langle x_1, \nabla\overline{\Psi}(x_1)\rangle - \langle x_1, x_0\rangle + 2(\Psi(x_0) + \overline{\Psi}(x_1) - \langle\nabla\overline{\Psi}(x_1), x_1\rangle) + \langle(\nabla\overline{\Psi}(x_1), x_1\rangle \\
&\quad - \langle x_0, \nabla\Psi(x_0)\rangle + \langle x_0, \nabla\Psi(x_0)\rangle - \langle x_0, x_1\rangle \\
&= 2[\Psi(x_0) + \overline{\Psi}(x_1) - \langle x_0, x_1\rangle].
\end{aligned}
$$

$\square$

Integrating equality (25) over the given transport plan $\pi$ and considering the formulas for the losses (4) and (16), we derive our Theorem 1.

**Proof of Theorem 1** (OFM and OT connection)

*Proof.* Main Integration Lemma 2 states that for any fixed points $x_0, x_1$ we have

$$\int_0^1 \|x_1 - x_0 - u_t^{\Psi}(x_t)\|^2 dt = 2[\Psi(x_0) + \overline{\Psi}(x_1) - \langle x_0, x_1\rangle].$$

Taking math expectation over any plan $\pi$ (integration w.r.t. points $x_0, x_1 \sim \pi$ ) gives

$$\underbrace{\mathbb{E}_{x_0,x_1\sim\pi}\int_0^1 \|u_t^{\Psi}(x_t) - (x_1 - x_0)\|^2 dt}_{=\mathcal{L}_{OFM}^{\pi}(\Psi)} = 2\cdot\underbrace{\mathbb{E}_{x_0,x_1\sim\pi}[\Psi(x_0) + \overline{\Psi}(x_1)]}_{=\mathcal{L}_{OT}(\Psi)} - \underbrace{2\cdot\mathbb{E}_{x_0,x_1\sim\pi}[\langle x_0, x_1\rangle]}_{=:\text{CONST}'(\pi)},$$

where $\text{CONST}'(\pi)$ does not depend on $\Psi$. Hence, both minimums of OFM loss $\mathcal{L}_{OFM}^{\pi}(\Psi)$ and of OT dual form loss $\mathcal{L}_{OT}(\Psi)$ are achieved at the same functions. $\square$

**Proof of Proposition 3** (Intractable Distance)

*Proof.* We recall the definitions of $\text{DIST}(u, u^*)$ (19) and FM loss $\mathcal{L}_{FM}^{\pi^*}(u)$ (10):

$$\text{DIST}(u, u^*) = \int_0^1 \int_{\mathbb{R}^D} \|u_t(x_t) - u_t^*(x_t)\|^2 \underbrace{\phi_t^* \# p_0(x_t)}_{=p_t^*(x_t)} dx_t dt,$$

$$\mathcal{L}_{FM}^{\pi^*}(u) = \int_0^1 \left\{ \int_{\mathbb{R}^D \times \mathbb{R}^D} \|u_t(x_t) - (x_1 - x_0)\|^2 \pi^*(x_0, x_1) dx_0 dx_1 \right\}, \; x_t = (1-t)x_0 + tx_1.$$

In the optimal plan $\pi^*$, each point $x_0$ almost surely goes to the single point $\nabla\Psi^*(x_0)$. Hence, in FM loss, we can leave only integration over initial points $x_0$ substituting $x_1 = \nabla\Psi^*(x_0)$ for fixed time $t$:

$$\int_{\mathbb{R}^D \times \mathbb{R}^D} \|u_t(x_t) - (x_1 - x_0)\|^2 \pi^*(x_0, x_1) dx_0 dx_1$$

$$= \int_{\mathbb{R}^D} \|u_t(x_t) - (\nabla\Psi^*(x_0) - x_0)\|^2 p_0(x_0) dx_0, \; x_t = (1-t)x_0 + t\nabla\Psi^*(x_0). \quad (37)$$

We notice that dynamic OT vector field $u^* = u^{\Psi^*}$ is the optimal one with potential $\Psi^*$. Moreover, for any point $x_t = (1-t)x_0 + t\nabla\Psi^*(x_0)$ generated by $u^*$, we can calculate $u_t^*(x_t) = u_t^{\Psi^*}(x_t) = \nabla\Psi^*(x_0) - x_0$. It is the same expression as from (37), i.e.,

$$
\begin{aligned}
(37) &= \int_{\mathbb{R}^D} \|u_t(x_t) - (\nabla\Psi^*(x_0) - x_0)\|^2 p_0(x_0) dx_0 \\
&= \int_{\mathbb{R}^D} \|u_t(x_t) - u_t^*(x_t)\|^2 p_0(x_0) dx_0, \; x_t = (1-t)x_0 + t\nabla\Psi^*(x_0).
\end{aligned}
$$

Finally, we change the variable $x_0$ to $x_t = \phi_t^*(x_0)$, and probability changes as $p_0(x_0)dx_0 = \phi_t^* \# p_0(x_t)dx_t = p_t^*(x_t)dx_t$. In new variables, we obtain the result

$$\int\limits_{\mathbb{R}^D \times \mathbb{R}^D} \|u_t(x_t) - (x_1 - x_0)\|^2 \pi^*(x_0, x_1)dx_0 dx_1 = \int_{\mathbb{R}^D} \|u_t(x_t) - u_t^*(x_t)\|^2 p_t^*(x_t)dx_t.$$

Hence, the integration over time $t$ gives the desired equality

$$\text{DIST}(u, u^*) = \mathcal{L}_{FM}^{\pi^*}(u),$$

and $\mathcal{L}_{FM}^{\pi^*}(u^*) = \text{DIST}(u^*, u^*) = 0.$ $\qquad\square$

**Proof of Proposition 4** (Tractable Distance For OFM)

*Proof.* For the vector field $u^\Psi$, we apply the formula for intractable distance from Proposition 3, i.e,

$$\text{DIST}(u^\Psi, u^{\Psi^*}) = \mathcal{L}_{FM}^{\pi^*}(u^\Psi) - \mathcal{L}_{FM}^{\pi^*}(u^{\Psi^*}) \overset{(16)}{=} \mathcal{L}_{OFM}^{\pi^*}(\Psi) - \mathcal{L}_{OFM}^{\pi^*}(\Psi^*).$$

According to Main Integration Lemma 2, for any plan $\pi$ and convex function $\Psi$, we have equality

$$\underbrace{\mathbb{E}_{x_0, x_1 \sim \pi} \int_0^1 \|u_t^\Psi(x_t) - (x_1 - x_0)\|^2 dt}_{=\mathcal{L}_{OFM}^\pi(\Psi)} = 2 \cdot \underbrace{\mathbb{E}_{x_0, x_1 \sim \pi}[\Psi(x_0) + \overline{\Psi}(x_1)]}_{=\mathcal{L}_{OT}(\Psi)} - \underbrace{2 \cdot \mathbb{E}_{x_0, x_1 \sim \pi}[\langle x_0, x_1 \rangle]}_{=:\text{CONST}'(\pi)}.$$

Since $\text{CONST}'(\pi)$ does not depend on $\Psi$, we have the same constant with $\Psi = \Psi^*$ and can eliminate it, i.e.,

$$\begin{cases} \mathcal{L}_{OFM}^\pi(\Psi) = 2 \cdot \mathcal{L}_{OT}(\Psi) - \text{CONST}'(\pi), \\ \mathcal{L}_{OFM}^\pi(\Psi^*) = 2 \cdot \mathcal{L}_{OT}(\Psi^*) - \text{CONST}'(\pi) \end{cases}$$
$$\Downarrow$$
$$\mathcal{L}_{OFM}^\pi(\Psi) - \mathcal{L}_{OFM}^\pi(\Psi^*) = 2 \cdot \mathcal{L}_{OT}(\Psi) - 2 \cdot \mathcal{L}_{OT}(\Psi^*). \tag{38}$$

The right part of (38) does not depend on a plan $\pi$, thus, the left part is invariant for any plan including optimal plan $\pi^*$, i.e.,

$$\mathcal{L}_{OFM}^\pi(\Psi) - \mathcal{L}_{OFM}^\pi(\Psi^*) = \mathcal{L}_{OFM}^{\pi^*}(\Psi) - \mathcal{L}_{OFM}^{\pi^*}(\Psi^*) = \text{DIST}(u^\Psi, u^{\Psi^*}).$$
$\qquad\square$

# B Experiments details

## B.1 OFM implementation

To implement our proposed approach in practice, we adopt fully-connected ICNN architectures proposed in [30, Appendix B2] (`W2GN_ICNN`) and [28, Appendix E1] (`CPF_ICNN`). To ensure the convexity, both architectures place some restrictions on the NN's weights and utilized activation functions, see the particular details in the corresponding papers. We take the advantage of their official repositories:

We aggregate the hyper-parameters of our Algorithm 1 and utilized ICNNs for different experiments in Table 3. In all our experiments as the $SubOpt$ optimizer we use LBFGS (`torch.optim.LBFGS`) with $K_{\text{sub}}$ optimization steps and early stopping criteria based on gradient norm. To find the initial point $z_0^i$ (Step 5 of our Algorithm 1), we initialize $SubOpt$ with $x_{t^i}^i$. As the $Opt$ optimizer we adopt Adam with learning rate $lr$ and other hyperparameters set to be default.

**Minibatch.** Similarly to OT-CFM, in some of our experiments we use non-independent initial plans $\pi$ to improve convergence. We construct $\pi$ as follows: for independently sampled minibatches $X_0, X_1$ of the same size $B$, we build the optimal discrete map and apply it to reorder the pairs of samples. We stress that considering minibatch OT for our method is done exclusively to speed up the training process. Theoretically, our method is agnostic to initial plan $\pi$ and is guaranteed to have an optimum in dynamic OT solution.

| Experiment | ICNN architecture $\Psi_\theta$ | $K$ | $B$ | $lr$ | $K_{\text{sub}}$ |
|---|---|---|---|---|---|
| Illustrative 2D | `CPF_ICNN`, $\mathbb{R}^2 \to \mathbb{R}$, Softplus, [1024, 1024] | 30K | 1024 | $10^{-2}$ | 5 |
| W2 bench., dim. $D$ | `W2GN_ICNN`, $\mathbb{R}^D \to \mathbb{R}$, CELU, [128, 128, 64] | 30K | 1024 | $10^{-3}$ | 50 |
| ALAE | `W2GN_ICNN`, $\mathbb{R}^{512} \to \mathbb{R}$, CELU, [1024, 1024] | 10K | 128 | $10^{-3}$ | 10 |

Table 3: Hyper-parameters of our OFM solvers in different experiments

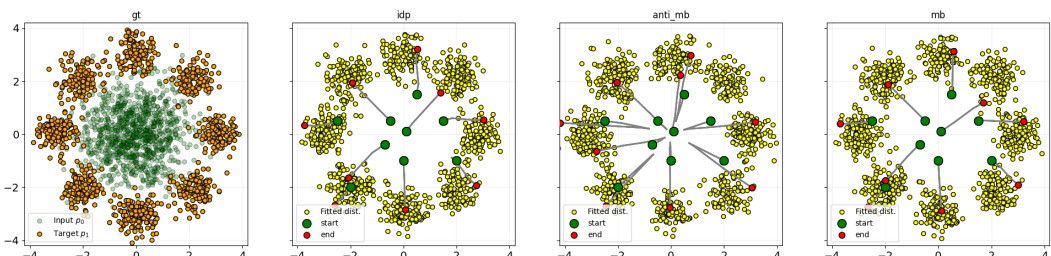

Figure 8: Performance of Flow Matching on *Gaussian→Eight Gaussians* 2D setup.

## B.2    2D Example: Comparison with Flow Matching

In this subsection, we illustrate that the restriction of the optimization domain only to optimal vector fields in FM loss is crucial for the plan independency and straightness of the obtained trajectories.

For that, we run the same setup from Section 4.1 but with vanila Flow Matching instead of OFM. The obtained trajectories and learned distributions for different initial plans are depicted in Figure 8.

In comparison with our OFM (Figure 6), basic FM yields more curved trajectories, especially with the misleading anti-minibatch plan. The learned distributions for all plans are similar to the target.

## B.3    Benchmark details

**Metrics.** Following the authors of the benchmark [33], to assess the quality of retrieved transport map $T$ between $p_0$ and $p_1$, we use *unexplained variance percentage* (UVP): $\mathcal{L}^2\text{-UVP}(T) := 100 \cdot \|T - T^*\|^2_{\mathcal{L}^2(p_0)}/\text{Var}(p_1)\%$. For values $\mathcal{L}^2-\text{UVP}(T) \approx 0\%$, $T$ approximates $T^*$, while for values $\geq 100\%$ $T$ is far from optimal. We also calculate the *cosine similarity* between ground truth directions $T^* - \text{id}$ and obtained directions $T - \text{id}$, i.e.,

$$\cos(T - \text{id}, T^* - \text{id}) = \frac{\langle T - \text{id}, T^* - \text{id}\rangle_{\mathcal{L}^2(p_0)}}{\|T - \text{id}\|_{\mathcal{L}^2(p_0)} \cdot \|T^* - \text{id}\|_{\mathcal{L}^2(p_0)}} \in [-1, 1].$$

For good approximations the cosine metric is approaching 1. We estimate $\mathcal{L}^2-\text{UVP}$ and $\cos$ metrics with $2^{14}$ samples from $p_0$.

In the experiments, we use the exponential moving average (EMA) [45, 24] of the trained model weights. EMA creates a smoothed copy of the model whose weights are updated at each new training iteration $t + 1$ as $\theta^{\text{ema}}_{t+1} = \alpha\theta^{\text{ema}}_t + (1 - \alpha)\theta_{t+1}$, where $\theta_{t+1}$ are the newly updated original trained weights. We calculate final metrics with $\alpha = 0.999$.

Solvers' results for $\cos$ metric are presented in Table 4.

| Solver | Solver type | $D=2$ | $D=4$ | $D=8$ | $D=16$ | $D=32$ | $D=64$ | $D=128$ | $D=256$ |
|---|---|---|---|---|---|---|---|---|---|
| MMv1* [54] | Dual OT solver | 0.99 | 0.99 | 0.99 | 0.99 | 0.98 | 0.99 | 0.99 | 0.99 |
| Linear* | Baseline | 0.75 | 0.80 | 0.73 | 0.73 | 0.76 | 0.75 | 0.77 | 0.77 |
| OT-CFM MB [55] | | 0.999 | 0.985 | 0.978 | 0.968 | 0.975 | 0.96 | 0.949 | 0.915 |
| RF [40] | | 0.87 | 0.75 | 0.65 | 0.67 | 0.72 | 0.70 | 0.70 | 0.70 |
| *c*-RF [39] | Flow Matching | 0.989 | 0.83 | 0.83 | 0.78 | 0.778 | 0.762 | 0.748 | 0.73 |
| OFM Ind (**Ours**) | | 0.999 | 0.993 | 0.993 | 0.993 | 0.999 | 0.966 | 0.992 | 0.981 |
| OFM MB (**Ours**) | | **0.999** | **0.994** | **0.995** | **0.994** | **0.999** | **0.970** | **0.994** | **0.986** |

Table 4: $\cos$ values of solvers fitted on high-dimensional benchmarks in dimensions $D = 2, 4, 8, 16, 32, 64, 128, 256$. The best metric over *Flow Matching based* solvers is **bolded**. * Metrics for MMv1 and linear baseline are taken from [33].

**Details of Solvers.** Neural networks' architectures of competing Flow Matching methods and their parameters used in benchmark experiments are presented in Table 5. In this Table, "FC" stands for "fully-connected".

| Solver | Architecture | Activation | Hidden layers | Optimizer | Batch size | Learning rate | Iter. per round * rounds |
|---|---|---|---|---|---|---|---|
| OT CFM [55] | FC NN $\mathbb{R}^D \times [0, 1] \to \mathbb{R}^D$ | ReLU | [128, 128, 64] | RMSprop | 1024 | $10^{-3}$ | 200.000 |
| RF [40] | FC NN $\mathbb{R}^D \times [0, 1] \to \mathbb{R}^D$ | ReLU | [128, 128, 64] | RMSProp | 1024 | $10^{-4}$ | 65.000 * 3 |
| $c$-RF [39] | FC NN $\mathbb{R}^D \times [0, 1] \to \mathbb{R}$ | ReLU | [128, 128, 64] | RMSProp | 1024 | $10^{-5}$ | 100.000 * 2 |

Table 5: Parameters of models fitted on benchmark in dimensions $D = 2, 4, 8, 16, 32, 64, 128, 256$.

Time variable $t$ in $(c-)$RF and OT-CFM's architectures is added as one more dimensionality in input without special preprocessing. In RF and $c$-RF, ODE are solved via Explicit Runge-Kutta method of order $5(4)$ [16] with absolute tolerance $10^{-4} - 10^{-6}$. In OFM and $c$-RF, gradients over input are calculated via autograd of `PyTorch`.

Following the authors of RF [40], we run only $2 - 3$ rounds in RF. In further rounds, straightness and metrics change insignificantly, while the error of target distribution learning still accumulates.

Our implementations of OT-CFM [55] and RF [40] are based on the official repositories:



https://github.com/atong01/conditional-flow-matching
https://github.com/gnobitab/RectifiedFlow



Implementation of $c$-RF follows the RF framework with the modification of optimized NN's architecture. Instead of $\mathbb{R}^D \times [0, 1] \to \mathbb{R}^D$ net, we parametrize time-dependent scalar valued model $\mathbb{R}^D \times [0, 1] \to \mathbb{R}$ which gradients are set to be the vector field.

## B.4 Unpaired Image-to-image transfer details

To conduct the experiments with high-dimensional I2I translation empowered with pretrained ALAE autoencoder, we adopt the publicly available code:



https://github.com/SKholkin/LightSB-Matching.



Additional qualitative results for our method are provided in Figure 9.

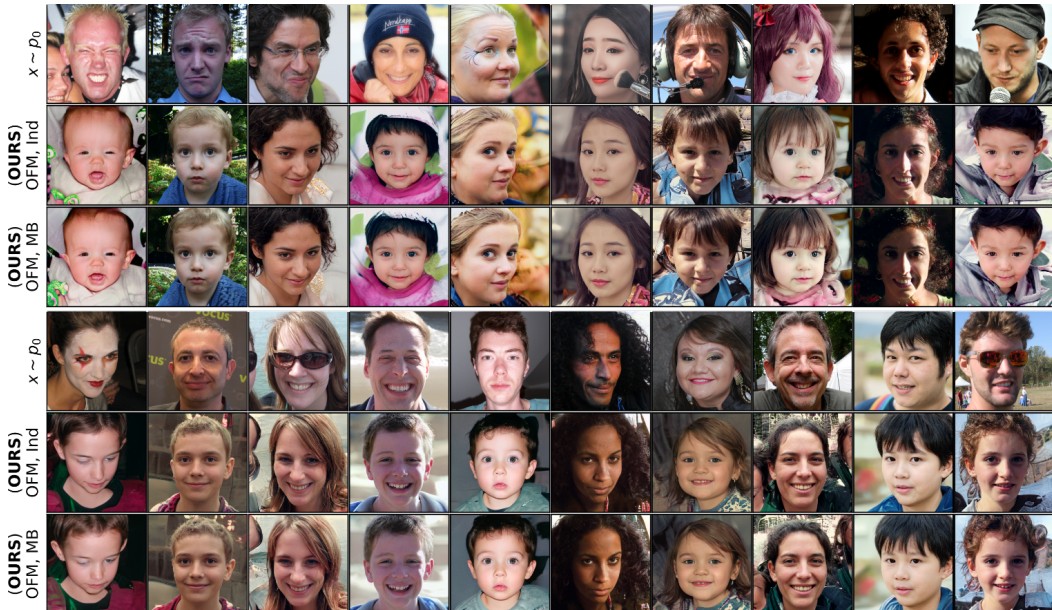

Figure 9: Unpaired I2I *Adult→Child* by **our** OFM solver, ALAE $1024 \times 1024$ FFHQ latent space. The samples are uncurated.

### B.5 Computation time

In what follows, we provide approximate running times for training our OFM and other FM-based method in different experiments with hyper-parameters provided in Table 3.

In the Illustrative 2D experiment, the training takes $\approx 1.5$ hours on a single 1080 ti GPU. In the Wasserstein-2 benchmark, the computation time depends on the dimensionality $D = 2, 4, \ldots, 256$. Totally, all the benchmark experiments (both with Ind and MB plan $\pi$) take $\approx 3$ days on three A100 GPUs. In the ALAE experiment, the training stage lasts for $\approx 5$ hours on a single 1080 ti GPU.

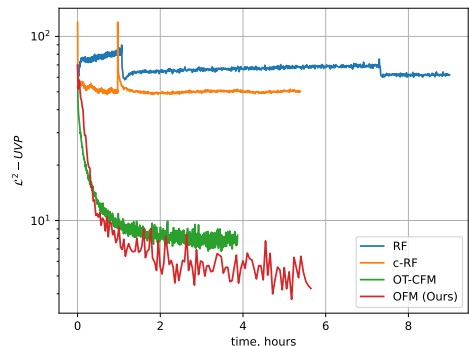

Figure 10: $\mathcal{L}^2-$UVP metric depending on the elapsed training time in dimension $D = 32$.

For better understanding of methods' behaviour over time, we depict achieved $\mathcal{L}^2-$UVP metric on the benchmark ($D = 32$) depending on elapsed training time in Figure 10. We note that the training iteration of OFM is computationally expensive, but it requires less steps to achieve the best results.

### B.6 Amortization technique

In order to train our OFM, we need to efficiently solve subproblem (17). As an example of more advanced technique rather than LBFGS solver, we discuss amortization trick proposed in [2].

Namely, we find an approximate solution of (17) at point $x_t$ and time $t$ with an extra MLP $A_\phi(\cdot, \cdot)$ : $\mathbb{R}^D \times [0, 1] \to \mathbb{R}^D$ :

$$A_\phi(x_t, t) \approx \arg \min_{z_0 \in \mathbb{R}^D} \left[ \frac{(1-t)}{2} \|z_0\|^2 + t\Psi(z_0) - \langle x_t, z_0 \rangle \right], \tag{39}$$

and then run sub-problem solver (LBFGS) initialized with $A_\phi(x_t, t)$ until convergence. We modify the training pipeline and include learning of parameters $\phi$ of $A_\phi$ in Algorithm 2.

---

**Algorithm 2** Optimal Flow Matching with Amortization

---

**Input:** Initial transport plan $\pi \in \Pi(p_0, p_1)$, number of iterations $K$, batch size $B$, optimizer $Opt$, amortization optimizer $AmorOpt$, sub-problem optimizer $SubOpt$, ICNN $\Psi_\theta$, MLP $A_\phi$

1: **for** $k = 0, \ldots, K - 1$ **do**
2:      Sample batch $\{(x_0^i, x_1^i)\}_{i=1}^B$ of size $B$ from plan $\pi$;
3:      Sample times batch $\{t^i\}_{i=1}^B$ of size $B$ from $U[0, 1]$;
4:      Calculate linear interpolation $x_{t^i}^i = (1 - t^i)x_0^i + t^i x_1^i$ for all $i \in \overline{1, B}$;
5:      Compute initialization $z_{init}^i = A_\phi(x_{t^i}^i, t^i)$ for all $i \in \overline{1, B}$;
6:      Find detached solution $z_0^i$ of (17) via $SubOpt$ initialized with $z_{init}^i$ for all $i \in \overline{1, B}$;
7:      Calculate OFM loss $\hat{\mathcal{L}}_{OFM}$

$$\hat{\mathcal{L}}_{OFM} = \frac{1}{B} \sum_{i=1}^B \left\langle \text{NO-GRAD} \left\{ 2 \left( t^i \nabla^2 \Psi_\theta(z_0^i) + (1 - t^i)I \right)^{-1} \frac{(x_0^i - z_0^i)}{t^i} \right\}, \nabla \Psi_\theta(z_0^i) \right\rangle;$$

8:      Update parameters $\theta$ via optimizer $Opt$ step with $\frac{d\hat{\mathcal{L}}_{OFM}}{d\theta}$;
9:      Calculate Amortization loss $\mathcal{L}_{Amor}$

$$\mathcal{L}_{Amor} = \frac{1}{B} \sum_{i=1}^B \|z_{init}^i - z_0^i\|^2;$$

10:      Update parameters $\phi$ via optimizer $AmorOpt$ step with $\frac{d\mathcal{L}_{Amor}}{d\phi}$;
11: **end for**

---

During the experiments, we did not find any improvements of the final metrics, in comparison with the original OFM with the same hyperparameters. However, this augmentation potentially can cause a shrinking of the overall training time. During training, $A_\phi(\cdot, \cdot)$ learns to predict more and more accurate initial solution $z_{init}^i$ and, thus, reduces the required number of the expensive $SubOpt$ steps.

## C    Limitations and Broader Impact

We find 3 limitations of our OFM which are to be addressed in the future research.

**(a) Flow map inversion.** During training, we need to compute $(\phi_t^{\Psi_\theta})^{-1}(\cdot)$ via solving strongly convex subproblem (17). In practice, we approach it by the standard gradient descent (with LBFGS optimizer), but actually there exist many improved methods to solve such conjugation problems more effectively in both the optimization [58, 26] and OT [2, 43]. This provides a dozen of opportunities for improvement, but leave such advanced methods for future research.

**(b) ICNNs.** It is known that ICNNs may underperform compared to regular neural networks [33, 31]. Thus, ICNN parametrization may limit the performance of our OFM. Fortunately, deep learning community actively study ways to improve ICNNs [10, 8, 49, 27] due to their growing popularity in various tasks [62, 37, 11]. We believe that the really expressive ICNN architectures are yet to come.

**(c) Hessian inversion.** We get the gradient of our OFM loss via formula from Proposition 1. There we have to invert the hessian $\nabla^2 \Psi(\cdot)$, which is expensive. We point to addressing this limitation as a promising avenue for future studies.

---

**Broader impact**. This paper presents work whose goal is to advance the field of Machine Learning. There are many potential societal consequences of our work, none of which we feel must be specifically highlighted here.

# D  Check List

