# OpenReview forum: "Optimal Flow Matching: Learning Straight Trajectories in Just One Step"
_NeurIPS.cc/2024/Conference — NeurIPS 2024 poster_

### Official Review · Reviewer_qt8g · 2024-06-29

**Soundness:** 4
**Presentation:** 3
**Contribution:** 4
**Rating:** 7
**Confidence:** 3

**Summary:**

This paper aims to solve the optimal transport problem between two distributions within a flow matching framework.
This can be achieved by finding the best velocity model $u_t$ in a class of optimal vector fields, i.e., $u_t$ be implicitly induced by defining the $x_1=\nabla \Phi(x_0)$ where $\Phi$ is a convex function.
The authors prove that the resulting (optimal) flow matching loss is equivalent to the dual formulation of the OT problem and regression towards the dynamic OT field.
However, optimizing the optimal flow matching loss can be computationally expensive, as the velocity model is only implicitly defined.
Despite this, the study of the relationship between the optimal flow matching problem and optimal transport can be a strong contribution.

**Strengths:**

- In general, this paper is well-written and easy to follow.

- Lemma 1 and its corollaries make a strong contribution to the community. It proves that the OT problem can be equivalently solved by simply optimizing the flow matching loss in the ``optimal vector fields'' family. In fact, this is a rather surprising result for me.

- In theory, the proposed optimal flow matching method can learn the OT map after one round of training, regardless of the reference distribution.

- The experiments are well-designed and convincing. The 2D experiment with different reference distributions (especially the anti-minibatch) is illustrative.

**Weaknesses:**

- The practical implementation of OFM is rather computationally expensive. It would be fine to add a table to directly compare the training times of different methods, or a figure showing the convergence as a function of computation time. I don't think this will lessen the contribution of this paper.

- As the authors claim that OFM learns straighter trajectories, comparing OFM to other baselines on standard generation benchmarks, e.g., CIFAR10, would be better.

**Questions:**

On the image-to-image transfer task, do you use unpaired training data? If the answer is yes, is it costly to compute the minibatch OT? Still, in this case, what is the tensor shape of the 512-dimensional latent space? I think 512 is much smaller than other latent space models.

**Limitations:**

the authors adequately addressed the limitations

---

> ### Author Rebuttal · Authors · 2024-08-07
>
> Dear reviewer, thank you for your feedback and kind words. Please find below the answers to your questions.
>
> **(1) Convergence time. Plots of convergence.**
>
> Following your request, we provide the plots of convergence (in $L^2$-UVP) depending on training time in the experiment on the W2 benchmark (Section 4.2) in dimension $D=32$. Please see **Figure 3 in the PDF file**. OFM iteration takes more time than iteration of OT-CFM, RF or $c-RF$ during the first FM round. After the first FM round, RF and $c-$RF slow down considerably and lose to OFM. Fortunately, OFM requires much fewer iterations to achieve the best metrics and outperforms all other methods in the first hours of the training.
>
> We will include this requested plot to the appendix of the final version of the paper.
>
>
> **(2) Comparing OFM to other baselines on standard generation benchmarks.**
>
> In theory, OFM can work directly in pixel-space (e.g., of CIFAR10) and has no limitation for dimensionality. However, in practice, the ML community has not yet designed input convex neural networks (ICNN) architectures that work well in the pixel-space (see, e.g., [1] for reference). Hence, for now image generation probably remains infeasible, and we encourage the ML community to invest into developing ICNN architectures.
>
>
>
> **(3) On the image-to-image transfer task, do you use unpaired training data? Is it costly to compute the minibatch OT? Still, in this case, what is the tensor shape of the 512-dimensional latent space? I think 512 is much smaller than other latent space models.**
>
>
>
> Yes, we use unpaired data. The latent space of ALAE autoencoder has dimension $D=512$ (e.g., the full encoded dataset has size $N\times 512$, where $N$ is the number of samples). Discrete OT computation in $512$-dimensional space actually is not very costly, since this task is much easier than the conjugation or hessian inversion used in training. OT computation requires $O(D\cdot B^3)$ time, where $B$ is the batch size (it is usually considerably smaller than dimension $D$). For comparison, the hessian inversion itself takes $O(BD^3)$.
>
>
>
> **Concluding remarks for all questions.**  We thank the reviewer one more time for high evaluation of our work. Please respond to our post to let us know if the clarifications above suitably address your concerns about our work. We are happy to address any remaining points during the discussion phase.
>
> **References.**
>
> [1] Korotin, Alexander, et al. "Do neural optimal transport solvers work? a continuous Wasserstein-2 benchmark." Advances in neural information processing systems 34 (2021): 14593-14605.

---

> > ### Comment · Reviewer_qt8g · 2024-08-08
> >
> > I thank the authors for their clarifications and additional experiments. I will keep my score.

---

### Official Review · Reviewer_nvT5 · 2024-07-12

**Soundness:** 2
**Presentation:** 2
**Contribution:** 2
**Rating:** 5
**Confidence:** 4

**Summary:**

This work proposes Optimal Flow Matching (OFM), which restricts standard Flow Matching to specific vector fields that yield straight trajectories by design. This is implemented by considering vector fields such that a convex function exists whose gradient pushes the initial to the final point.

The authors propose to solve this optimization problem with training that involves solving a convex optimization problem in each iteration to obtain the inverse of the flow map. Then, it is shown that this procedure has the same minimizer as the one minimizing the dual OT loss. The authors empirically validate OFM quantitatively on the Wasserstein-2 benchmark and qualitatively on FFHQ.

**Strengths:**

- The authors derive a new training procedure based on Flow Matching restricted to fully straight vector fields implemented through the gradient of a convex function.
- The authors provide a new insight into the connection between minimizing an OT loss and Flow Matching. Specifically, they show that the minimizer of these two losses is equivalent when restricted to straight vector fields.

**Weaknesses:**

Missing comparison and contextualization of related work:
- In [1], it was proposed to approximate the computation of the conjugate with amortized optimization and solve this with Adam or L-BFGS. The proposed *SubOpt* in this work (solved with Adam/L-BFGS) is equivalent to the convex conjugate optimization when $t=1$.  A discussion between these two optimization schemes should be included. Specifically, how does OFM compare to [1] when only sampling $t=1$? What is gained by considering $t\sim \mathcal{U}[0,1]$? Additionally, the Wasserstein-2 benchmark results from [1] are not considered in the paper. This is the most related method and outperforms the method proposed in the paper on all dimensions. These results should be included as well as a comparative discussion.

Experiments:
- No quantitative results are provided for the unpaired image-to-image translation experiments. It is hard to generalize the results over a few qualitative results. Thus, including some commonly used quantitative metrics like FID or the ones used in the Wasserstein-2 benchmark on CelebA64, as done in [1], would be very beneficial.

Minor Weaknesses:
- Inconsistency on the optimizer used to solve *SubOpt*. In Appendix C, it is stated that "in all our experiments as the subOpt optimizer we use LBFGS"[line 557] while in Appendix B, it says "(with Adam optimizer)"[line 534]. Which one is actually used? How do Adam and LBFGS compare?
- Some of the wording can be improved, e.g. "... via fancy formula (18) is not trivial and requires tricky integration techniques."[line 194]

[1] Brandon Amos. "On amortizing convex conjugates for optimal transport". In ICLR 2023.

**Questions:**

- The goal of OFM is to learn a time-independent one-step model that translates $x_0$ to $x_1$. Then, during inference, no ODE is solved. As mentioned above, some more motivation could improve clarity here. Specifically, explaining why it is beneficial to formulate this in the FM framework considering $t\sim \mathcal{U}[0,1]$. Would the proposed method also work when only sampling $t=1$? The authors only mention that the subproblem (20) is $(1-t)$-strongly convex, which gives a benefit compared to $t=1$. Are there other benefits of the proposed formulation?
- In Figure 2, it is shown that OFM learns the same solution for different choices of couplings. In [2], it was shown that only the OT coupling is preserved when using standard Flow Matching. What do these illustrative examples look like for standard FM training? I assume they would learn different trajectories, but do the learned endpoints differ?
- In this work, only ICNNs are considered. As shown in previous works, even when wanting to learn a convex function, MLPs can also work well, i.e., in [1], they work better compared to ICNNs. Does OFM also work with standard MLPs/CNNs/etc? How is the comparative performance?

[1] Brandon Amos. "On amortizing convex conjugates for optimal transport". In ICLR 2023.

[2] Valentin De Bortoli and Guan-Horng Liu and Tianrong Chen and Evangelos A. Theodorou and Weilie Nie. "Augmented Bridge Matching". In Arxiv 2023.

**Limitations:**

- The limitations of the flow map inversion, the Hessian computation, and ICNNs are laid out in the paper, and details for the experiments' computational time are provided.
- While these computational times give some insight, detailed information on time and space complexity with respect to the input dimension would be even more beneficial. Additionally, comparing the runtime of OFM and standard FM training would provide a more illustrative comparison of raw computational time.

---

> ### Author Rebuttal · Authors · 2024-08-07
>
> Dear reviewer, thank you for your detailed feedback. Please find below the answers to your questions.
>
> **(1)  The reduction of our OFM to sampling of $t = 1$ only.**
>
> Our method requires considering all times $t\in [0,1]$ by its design. Considering $t=1$ *breaks the theory and does not work*. Indeed, this can be seen both theoretically and practically. In case $t=1$, loss (17) in OFM becomes
> $$\min\limits_{\Psi \text{ convex}}\int\limits_{\mathbb{R}^D \times \mathbb{R}^D} \left|\left|\nabla \overline{\Psi} (x_1) - x_0 \right|\right|^2 \pi(x_0, x_1) dx_0 dx_1.$$
>
> Without $t\sim U[0,1]$, loss becomes **plan-dependent**, and OT recovery is not guaranteed. Hence, we lose both the desired connections with FM and OT.
> To also confirm this empirically, we run this optimization on a 2D Gaussian to Mixture of Gaussians setup for two plans: minibatch with 4 and 1024 samples in batch. The results are depicted in **Figure 1 in the PDF file**. One may clearly see that the learned map does not even match the target distribution.
> We also highlight that the objective above is **not** equivalent to the dual OT objective $\mathcal{L}_{OT}$ (equation (4) of our paper), which involves the potential $\Psi$ and its conjugate $\overline{\Psi}$.
>
> **(2) Benefits of the dynamic formulation ($t\in [0,1]$) over the static one (in addition to the strong convexity).**
>
>
> The main reason to consider $t \sim U[0,1]$ is to pave a new bridge between two emerging subfields in deep learning: optimal transport and flow matching. We introduce novel methodology relating both of these problems. This development potentially allows to use the advantage of one field to improve the methods in the other one. And, indeed, stronger convexity is a good benefit of the dynamic loss, but it is a bonus, not an initial goal.
>
> **(3) Comparison/relation to the dual OT optimization approach [1].**
>
>
> We primarily aimed to compare *flow matching-based methods* with each other. Metrics from MMv1 are included for completeness, and they already give a general understanding about the quality and baselines of modern static OT methods. Method [1] surely demonstrates the better result, as it used advanced amortization techniques to optimize $\mathcal{L}_{OT}$. We will include the results of the method in the table.
>
> Overall, advanced amortization techniques such as [1] are orthogonal to our theoretical and methodological study and can be used on top of our method to further improve it. To provide the empirical evidence, we adapted the amortization scheme from [1] to our OFM. Namely, we find an approximate conjugation solution with an extra MLP $A_\phi(x_t,t):\mathbb{R}^D \times [0,1] \to \mathbb{R}^D $:
> \begin{eqnarray}
> A_\phi(x_t, t) \approx \arg\min_{z_0 \in \mathbb{R}^D} \left[ \frac{(1-t)}{2} \| z_0\|^2 + t \Psi (z_0) - \langle x_t , z_0 \rangle \right],
> \end{eqnarray}
> and repeat the training pipeline from [1] but with an extra condition on $t$. We run an additional OFM experiment on $D = 32$ with amortization and regression loss. This augmentation gives a boost for metrics in comparison with basic minimization techniques, see **Figure 4 in the PDF file**. In the final version of the paper, we will add an appendix about the amortization technique for OFM.
>
> **(4) Inconsistency on the optimizer used to solve SubOpt. In Appendix C, it is stated that "in all our experiments as the subOpt optimizer we use LBFGS"[line 557] while in Appendix B, it says "(with Adam optimizer)"[line 534]. Which one is actually used? How do Adam and LBFGS compare?**
>
>
> We are sorry for this inconsistency in our text. In all our experiments, we used the LBFGS optimizer. We kindly propose you to take a look at our code. We will fix the misprint in Appendix B in the final version. Regarding Adam, in our early experiments we indeed experimented with this optimizer, but eventually found that LBFGS works better (it is faster when achieving the same tolerance). We humbly think that the detailed comparison between the optimizers is out of scope of our work.
>
>
> **(5) ICNNs are considered. Does OFM also work with standard MLPs/CNNs, etc.?**
>
> Following your request, we conducted several additional experiments with MLP parameterization of $\Psi$ instead of ICNN. Unfortunately, they failed to converge. Probably, this happens due to violation of conjugated functions' properties for non-convex functions required in OFM. We leave studying this phenomenon for future works.
>
>
>
> **(6) Quantitative results are provided for the unpaired image-to-image translation experiments.**
>
>
> Following your request, we calculated FID for I2I experiments. See our General response section and **Table 1 from the attached pdf**. According to the results, our OFM method beats all the baselines.
>
>
> **(7) How do the illustrative 2D examples look like for standard FM training with ID, MB, anmiMB plans? Do the endpoints/trajectories differ?**
>
>
> Following your request, we run experiments with FM under independent, minibatch and anti-minibatch plans on 2D example, see Figure 2 in the PDF file. Unlike OFM with straight paths, FM trajectories differ a lot depending on plan, curvature increases in order "mb", "idp" and "anti-mb". We, in particular, highlight the high curvature of antimibatch's trajectories. The generated distributions of end points and the target distribution are close in all the cases.
>
> **Concluding remarks for all questions.**  We thank the reviewer one more time for the interesting and important questions. Please respond to our post to let us know if the clarifications and additional experiments above suitably address your concerns about our work. We are happy to address any remaining points during the discussion phase; if the responses above are sufficient, we kindly ask that you consider raising your score.

---

> > ### Comment · Reviewer_nvT5 · 2024-08-12
> >
> > Thank you for the detailed comments and added experiments. This cleared up a lot of my questions, and I've decided to raise my score 4->5 as I think the paper has an interesting theoretical contribution.
> >
> > I still have concerns regarding the scalability of the method because of its reliance on ICNNs making it very hard to scale up to real word examples like pixel-space image generative modeling. The authors mention that the main competitive methods of their proposed method are different Flow Matching methods but the work does not include results on any common generative modeling benchmarks as also mentioned by Reviewer Bhuh. These other FM methods don't have any problems scaling up to arbitrary dimensions. This should be sufficiently discussed and contextualized in the revised manuscript. Also as mentioned in my rebuttal, I believe [1] is a more closely related work to the proposed method with respect to the Optimal Transport benchmark and outperforms it across all dimensions.

---

> > > ### Author Response · Authors · 2024-08-14
> > > **Thank you! and concluding comments**
> > >
> > > Dear Reviewer, we thank you for your answer and for highlighting the theoretical contribution of our work. We are very grateful for the score increase. Let us leave some concluding comments.
> > >
> > > 1. We agree that the reliance on the ICNNs makes the practical adaptation of our methodology to high-dim generative image setups to be tricky. Following your suggestions, we will expand the corresponding limitation subsection (**ICNNs**) accordingly.
> > >
> > > 2. The work [1] which is based on the amortization methodology will receive due attention in our revised manuscript. In particular, we acknowledge that we will include the benchmark results of [1] in our Table. With regard to the superior performance of [1], we humbly believe that achieving the SOTA metrics on the benchmark lies primarily on the surface of technical solutions. I.e., particular architectures, optimizers, schedulers, regularizers etc. To support our point, we provide the additional results for our method fitted on the benchmark, but with additional exponential moving average (EMA) of the trained model weights. Note that EMA does not change the training process itself - it just creates a smoothed copy of the model whose weights are updated (at each new training iteration $t + 1$) as $W^{\text{ema}}\_{t + 1} = \alpha W^{\text{ema}}\_{t} + (1 - \alpha) W\_{t + 1}$, where $W_{t + 1}$ are the newly updated original trained weights. Our results for $\mathcal{L}^2$-UVP are in the table below ($\alpha = 0.999$). For completeness, we also recall the metrics for our models without EMA and additionally grab the benchmark results from [1].
> > >
> > > | Solver/dim        | 2        | 4        | 8       | 16      | 32      | 64      | 128      | 256      |
> > > |-------------------|----------|----------|---------|---------|---------|---------|----------|----------|
> > > | OFM MB (ema)      |  $\textit{0.15}$   | $\textit{0.52}$   | $\textit{1.2}$   | $\textit{1.0}$   |  $\textbf{1.2}$ | 7.2     | $\textit{1.5}$    | 2.9      |
> > > | OFM MB            | 0.33     | 0.81     | 1.5     | 2.0     | 3.4     | 10      | 6.7      | 11.7     |
> > > | [1] (ICNNs) | 0.26     | 0.78     | 1.6     | 1.1     | $\textit{1.9}$     | $\textit{4.2}$   | 1.6      | $\textit{2.0}$    |
> > > | [1] (MLP)   | $\textbf{0.03}$ | $\textbf{0.22}$ |  $\textbf{0.6}$ |  $\textbf{0.8}$ | 2.0     |  $\textbf{2.1}$ |  $\textbf{0.67}$ |  $\textbf{0.59}$ |
> > >
> > > As we can see, the utilization of a rather cheap technique allows obtaining much better results, compared to those reported in our article. Regarding [1], it seems they did not utilize EMA in their experiments. Probably, the usage of EMA in their experiments also will lead to better results. Our newly provided table is not about competing with [1]. We just want to stress that the point of our benchmark experiments is to demonstrate the applicability and success of our proposed methodology - and we humbly think that we managed to achieve this goal - by demonstrating rather good and close-to-the-best metrics.
> > >
> > > One more time, we are grateful to the reviewer for the valuable comments, and we will incorporate the necessary edits in the revised manuscript.

---

> ### Author Response · Authors · 2024-08-12
> **The discussion period deadline is approaching...**
>
> Dear reviewer nvT5.
>
> Thank you for your time efforts to review our paper.
>
> The discussion period is coming to the end. We would appreciate your feedback on our responses. We are happy to answer any further questions during the remaining discussion period.
>
> Thank you in advance, Authors.

---

### Official Review · Reviewer_YPMs · 2024-07-13

**Soundness:** 2
**Presentation:** 1
**Contribution:** 2
**Rating:** 6
**Confidence:** 2

**Summary:**

The paper goal is to learn the optimal transport map between any two distributions. The approach  derives a flow matching loss which is minimized by the velocity field of the optimal transport map.

**Strengths:**

The paper investigate the interesting subject of the optimal transport problem.

**Weaknesses:**

The paper is written in an unprofessional form:
1. Theorems and propositions are structured in a way which makes it very hard to follow.

2. The author does not make any attempt to give an accessible main proof idea/proof sketch in the main paper. Specifically, it is expected to provide one for Theorem 1 which seems to be the key claim of the paper.

3. The proof of Theorem 1 in the appendix is unclear and unsatisfying.

4. The experiment section is rather minimal. In particular, it is not clear what Figure 3 in the Image-to-Image task presents.

**Questions:**

In essence, does the author claim that the $\mathcal{L}^{\pi}\_{OFM}(\Psi)$ loss presented in equation (16) is minimized by the velocity field of the optimal transport map for any plan $\pi$? As stated by the author, it seems this loss is by definition Flow Matching (FM) loss constrained to convex functions. For the independent plan, i.e., $\pi=p_0\times p_1$ the minimizer for the FM loss is explicitly given in [1],
$$ u_t(x) = \frac{1}{p_t(x)}\int u_t(x|x_1)\frac{p_t(x|x_1)p_1(x_1)}{p_t(x)}dx_1, $$
where for the linear interpolant $u_t(x|x_1) = \frac{x_1-x}{1-t}$. This minimizer is dependent solely on the source and target distribution (i.e., the plan for this case) and does not yield the optimal transport. If you could please explain how you overcame this.

[1] Lipman, Yaron, et al. "Flow matching for generative modeling." arXiv preprint arXiv:2210.02747 (2022).

**Limitations:**

-

---

> ### Author Rebuttal · Authors · 2024-08-07
>
> Dear reviewer, we thank you for your feedback. Please find below the answers to your questions.
>
> **(1) In essence, does the author claim that the  loss presented in equation (16) is minimized by the velocity field of the optimal transport map for any plan ? As stated by the author, it seems this loss is by definition Flow Matching (FM) loss constrained to convex functions.**
>
> Yes, we claim this. This is our main theoretical result presented in Section 3.1.
>
> **(2) For the independent plan, i.e.,  the minimizer for the FM loss is explicitly given in [1], where for the linear interpolant. This minimizer is dependent solely on the source and target distribution (i.e., the plan for this case) and does not yield the optimal transport. If you could please explain how you overcame this.**
>
> If we consider unconstrained FM loss minimization (w.r.t. the optimized vector field) as you suggest, the analytical solution is indeed given by the formula which you provided above. However, in our OFM, we propose to optimize FM loss only over **specific optimal vector fields**. In this case, **minimizers** of constrained and unconstrained problems are **not the same**. We theoretically prove that the minimizer in this case is the true OT vector field.
>
> **(3) The experiment section is rather minimal. In particular, it is not clear what Figure 3 in the Image-to-Image task presents.**
>
> The primal goal of the experimental section is to *empirically confirm our theoretical findings*. Namely, we check that our OFM always recovers the true OT map regardless of the input transport plan. We humbly believe that the section clearly demonstrates this:
>
> 1. In qualitative 2D-toy example of Section $4.1$ (Figure $2$), our OFM recovers the same trajectories for all input plan, *even for misleading anti-minibatch plan*.
>
> 2. In Section $4.2$, we quantitatively evaluate the ability to solve OT problems. We run our OFM, other FM-based methods and popular OT solver on OT Benchmark which knows ground truth OT solution. Then we directly compare obtained solutions with ground truth via L2-UVP and cosine metrics. We can see that our OFM outperforms other FM-based methods in recovering OT and demonstrates metrics *close to OT solvers*.
>
> Additionally, we conduct an illustrative experiment with image-to-image translation. Here the usage of OT is motivated by the necessity to keep the image content during the translation. In Figure $3$, we can see that OFM's results are the same for the both considered input plans and also the most plausible in comparison with adult images.
>
> As per request of the other reviewers, we also calculated FID on the I2I problem to quantitatively estimate the performance, see **Table 1 from the attached pdf**. Our OFM method outperforms all the baselines on this task.
>
> **(3) Theorems and propositions are structured in a way which makes it very hard to follow.**
>
> We are sorry that you found our paper hard to follow, but we understand that opinions on this topic differ, e.g., Reviewer qt8g, in contrast, found our paper well-written and easy to follow. To address your comment, we will restructure Section 3.1 in the final version. Specifically, we will split in to 2 subsections:
>
> - Subsection 3.1.1. (major) will contain **optimal vector fields, our proposed loss and direct statement of the main Theorem 1**. In turn, technical Lemma 1 will be moved to Appendix. This will help the reader to directly find the main result without any disctractions on technical things.
>
> - Subsection 3.1.2 (auxiliary) will contain various **supportive results**: Propositions 1-3 (loss reformulation, properties of the loss). Hence, the reader which is interested in the main result only will be able to easily skip this subsection and continue to the next (practical) section.
>
> *Could you please reply, does such a change suit you well?*
>
> **(4) The proof of Theorem 1 in the appendix is unclear and unsatisfying.**
>
> Please note that our main technical result is Lemma 1. It allows to take integral in FM loss over time $t$ and obtain analytical formula (18). The proof is based on rather non-trivial techniques from calculus and convex analysis: change of variables, line integrals, differentials and convex conjugation. We tried to do our best to make the proof accessible to the general ML community. Could you please point to particular aspects of the proof to clarify?
>
> In turn, our main Theorem 1 follows from Lemma 1. We take math expectation over plan $\pi$ from both sides of (18) and notice that the left part of (18) corresponds to OFM loss and the right part matches OT dual loss (4) up to constant.
>
> **(5) Accessible main proof idea/proof sketch in the main paper.**
>
> We do not include any sketch of the proofs because it is purely technical (but still very non-trivial) and heavily relies on doing calculus tricks such as the change of variables.
>
> In Lemma's proof (line 480), we firstly simplify OFM loss using Proposition 1. Then we make two changes of time variable $t$: $s = \frac{1}{t}, s' = \frac{1}{s-1}$, use eqn (15) and obtain integral in line 488. In line 489, we consider the curve $z_0(s')$ and take its differential $dz_0$ using implicit definition of $z_0(s')$ from eqn (15). Starting from here, we take line integrals over line $z_0$ to get the sum from (27).  Next we use closed forms of differentials, integrate it from $t = 0$ to $t = 1$ and obtain analytical sum (28). Finally, we use standard inversion and Fenchel-Young’s equalities to simplify the final sum.
>
> If you think a sketch of this type could be useful in the main text, we are happy to add it to the final version of the paper.
>
> **Concluding remarks for all questions.**  Please respond to our post to let us know if the clarifications above suitably address your concerns about our work. We are happy to address any remaining points during the discussion phase; if the responses above are sufficient, we kindly ask that you consider raising your score.

---

> ### Comment · Reviewer_YPMs · 2024-08-12
> **Reviewer response**
>
> I want to thank the authors for their response. Following the reviewers answers,  I have invested more time in better following the authors claim and proofs. It is now my understanding that the key results of the paper is that the minimizer of  FM loss constrained to velocity field which are a gradient of convex functions is the OT map. I acknowledged this is an important results and I will raise my score.

---

### Official Review · Reviewer_Bhuh · 2024-07-18

**Soundness:** 3
**Presentation:** 2
**Contribution:** 2
**Rating:** 5
**Confidence:** 4

**Summary:**

This paper introduces the Optimal Flow Matching (OFM) algorithm, which improves upon Rectified Flow and OT-CFM by generating exact straight trajectories and recovering the optimal transport map in one iteration. OFM optimizes Flow Matching loss using vector fields and a convex function. The algorithm is implemented with Input Convex Neural Networks (ICNNs) and includes an explicit gradient formula for loss calculation. Experiments demonstrate OFM's better performance on 2D, high-dimensional benchmarks,  and in unpaired image-to-image translation. OFM achieves good results in the latent space of a pretrained autoencoder without needing ODE integration.

**Strengths:**

Strengths:
- The problem is sound and essential.
- The proposal is supported by both theoretical and empirical evidence.

**Weaknesses:**

- The success of the algorithm in high-dimensional tasks relies on pretrained autoencoders. To what extent does the performance of the pretrained autoencoders affect the performance of the proposal?
- The assumption that the optimal transport map can be recovered in one iteration might not hold in all scenarios, particularly in more complex distributions.
- How do you calculate the initial transport plan? Does OFM suffer from distorted or biased dynamic OT solutions like OT-CFM?
- The proposal is compared with OT-CFM and Rectified flow. However, the main task of the baselines in their original paper was image generation in pixel space. Is it fair to exclude this task from the experiments?
- It is essential to provide more justification for using metrics in the experimental section. What are the advantages of these metrics over others in works on Unpaired Image-to-Image Translation tasks, such as [1] and [2]?

**Questions:**

- Does OFM still work in the pixel space or a higher-resolution dataset?
- Could you provide empirical running time and OFM time complexity?
- Does the method work for the conditional generation, which is feasible for Rectified flow?
- OFM does not need ODE for the sampling process; however, could using an ODE solver for sampling improve the performance of OFM, which is some kind of quality-time tradeoff?

**Limitations:**

**References**

[1] Zhu, Jun-Yan, et al. "Unpaired image-to-image translation using cycle-consistent adversarial networks." Proceedings of the IEEE international conference on computer vision. 2017.

[2] Xie, Shaoan, et al. "Unpaired image-to-image translation with shortest path regularization." Proceedings of the IEEE/CVF Conference on Computer Vision and Pattern Recognition. 2023.

---

> ### Author Rebuttal · Authors · 2024-08-07
>
> Dear reviewer, thank you for your feedback. Please find the answers to your questions and comments below.
>
> **(1) Pretrained autoencoders. Does OFM still work in the pixel space or a higher-resolution dataset?**
>
> In theory, OFM can work directly in pixel space and has no limitation for dimensionality. However, in practice, there are yet no input convex neural networks (ICNN) architectures that work well in the pixel-space (see, e.g., [1] for discussion). Hence, we use pretrained encoders to process images.
>
> We emphasize that this aspect is mostly related to ICNN rather than to our developed OFM Framework. We believe that our novel theoretical findings will inspire the ML community to develop new well-performing ICNNs that may work well in the pixel space.
>
> **(2) The proposal is compared with OT-CFM and Rectified Flow. However, the main task of the baselines in their original paper was image generation in pixel space. Is it fair to exclude this task from the experiments?**
>
> Please see the previous answer for the reason of excluding this type of experiments.
>
>
> **(3) The assumption that the optimal transport map can be recovered in one iteration might not hold in all scenarios, particularly in more complex distributions.**
>
> Dear reviewer, this is not an assumption. We provide and prove a *rigorous theoretical statement* (Theorem $1$) that our Optimal Flow Matching retrieves OT between **any** two absolutely continuous distributions in just one flow matching iteration. Moreover, our experiments on high-dimensional standard benchmark [1] in the field of OT (Table $1$) demonstrate that OFM actually finds a solution whose metrics are close to or better than those of the state-of-the-art OT approaches.
>
> **(4) Does OFM suffer from distorted or biased dynamic OT solutions like OT-CFM?**
>
> No, it does not. In our OFM, we run only one FM iteration with **any** transport plan as the initial plan (Theorem $1$) and provably recover the true OT map. That is, unlike OT-CFM, the initial plan theoretically *does not affect* the final solution (again, which is the OT map between distributions), i.e., there is no bias or distortion effect.
>
>
> **(4) How do you calculate the initial transport plan?**
>
> In 2D toy example (Section $4.1$), we practically demonstrate that OFM's final solutions do not depend on initial plans. In this setup, the considered plans are:
>
>  (a) Independent, where we independently sample batches from both distributions,
>
>  (b) Minibatch, where we independently sample batches and then rearrange them according to Discrete Optimal Transport (with $\ell^2$ cost) between them,
>
>  (c) Anti-Minibatch, where we independently sample batches and then rearrange them according to Discrete Optimal Transport based on **negative** Euclidean distance.
>
> The same approach with plans is used in the other experiments.
>
>
> **(4) It is essential to provide more justification for using metrics in the experimental section. What are the advantages of these metrics over others in works on Unpaired I2I Translation tasks?**
>
> The metrics which we use in Section 4.2 are specially designed for evaluation of the quality of the recovered OT solution in the benchmark experiment. These metrics are generally *not related to unpaired I2I problems*. Our goal there is to confirm that our OFM indeed learns OT. Specifically, we use the *unexplained variance percentage* (UVP) as the main metric: $\mathcal{L}^2$-UVP$(T) := 100 \cdot \| T - T^*\|^2_{\mathcal{L}^2(p_0)} / \text{Var}(p_1) \%$. It directly computes the squared error between ground truth OT map $T^{*}$ (which is known by the construction of the benchmark [1, Sec. 4.2]) and learned map $T$ and then normalizes the answer. Following the benchmark, we additionally compute cosine similarity between ground truth map directions and directions of $T.$
>
> Following your request, we additionally calculated FID for Unpaired Image-to-Image experiment from Section $4.3$. See our General response section and **Table 1 from the attached PDF**. According to the table, our method beats all the baselines.
>
>
> **(8) Does the method work for the conditional generation?**
>
> Unfortunately, ICNNs which we use in our method are not straightforward to adapt to conditional generation. In particular, to our knowledge, there are yet no research papers developing scalable ICNNs (especially, conditional) to work with pixel spaces. This is why we leave this interesting question for future research.
>
> **(9) Could using an ODE solver for sampling improve the performance of OFM?**
>
> If one uses ODE solver for the vector field generated by OFM, one gets the same solution as the one obtained by simply calculating the final point. It happens because OFM trajectories between the initial and final points are **perfectly** straight.
>
> **Concluding remarks.** We thank the reviewer one more time for the interesting and important questions. Please respond to our post to let us know if the clarifications above suitably address your concerns about our work. We are happy to address any remaining points during the discussion phase; if the responses above are sufficient, we kindly ask that you consider raising your score.
>
> **References**:
>
> [1] Korotin, Alexander, et al. "Do neural optimal transport solvers work? a continuous Wasserstein-2 benchmark." Advances in neural information processing systems 34 (2021): 14593-14605.

---

> > ### Comment · Reviewer_Bhuh · 2024-08-12
> >
> > Thank you the the authors for their responses. Your responses partially address my concerns about the paper. I have raised the score by one point to reflect my positive attitude toward the additional explanation and empirical results.

---

### Author Rebuttal · Authors · 2024-08-07

We thank the reviewers for their feedback and time. We appreciate that reviewers acknowledge: importance and soundness of the chosen problem (Bhuh, YPMs), our theoretical contribution connecting FM and OT methods (Bhuh, nvT5, qt8g), and experimental verification of the theoretical part (qt8g, Bhuh). Please find the answers to your questions in the respective posts below your reviews.

We additionally calculated FID for our considered Unpaired Image-to-Image problem to expand the experiments and make them more representative. The obtained results exhibit that our OFM approach quantitatively outperforms the considered flow-based baselines.

We attach the **PDF file** to this post with requested experiments containing:

- **(All Reviewers)** Table 1, FID metric for unpaired Image-to-Image real data problem. The considered methods are OFM, RF, c-RF and OT-CFM.

- **(Reviewer nvT5)** Figure 1, OFM setting $t \equiv 1$. The title "gt" stands for the source and target distributions. Titles "mb 64" and "mb 16" stand for minibatch plan with batchsize $64$ and $16$, respectively.

- **(Reviewer nvT5)** Figure 2, default FM with independent ("idp"), anti-minibatch ("anti mb") and minibatch ("mb") plans. The title "gt" stands for the source and target distributions.

- **(Reviewer qt8g)** Figure 3, convergence of L2-UVP loss depending on training time for OFM, FM, c-RF and OT-CFM methods.

- **(Reviewer nvT5)** Figure 4, OFM with and without amortization. The graph depicts L2-UVP metric during training.

---

> ### Author Response · Authors · 2024-08-14
>
> We are very grateful to the Reviewers for valuable questions, fruitful discussions and raising the scores. In the final version, we will make the following edits:
>
> 1) include time convergence graph (Appendix), extra OT baselines and OFM improved metrics (few rows in Table 1), I2I FID (Section $4.3$), FM examples and amortization (Appendix), (nvT5, qt8g);
>
> 2) restructure Section 3, (following the reply to YPMs);
>
> 3) discuss the intricacy of high-dimensional (i.e., image data) generation in Limitations section, (nvT5, Bhuh).

---

### Decision · Program_Chairs · 2024-09-25

**Decision:**

Accept (poster)

**Comment:**

This paper shows that the minimizer of the flow matching loss, where the velocity field is constrained to the form of W2-optimal velocity field (i.e., the velocity of the displacement map), is the actual Optimal Transport map. Based on this observation the author model the learnable velocity via the ICNN (convex functions) architecture and demonstrate the efficacy of their approach on a few 2D and image datasets, and comparing to other flow matching alternatives looking to find the Optimal Transport map.

Pros:
+ Reviewers found the main theoretical result in the paper an important one. In particular the use of Lemma 1 to prove main theorem.
+ New and interesting connection between Flow Matching and Optimal Transport.
+ A sufficient set of experiments supporting the theoretical findings.


Main reviewers' concerns included:
- Readability and lacking exposition of the main theoretical results > authors have addressed this and suggested a fix in final version.
- Unclear relation to SubOpt > authors clarified their method requires optimizing over all times $t\in [0,1]$.
- Scalability issues of ICNN, computational overhead of the method, convergence times > authors will address this in limitation and/or experiments and/or appendix section in the revised version.

Overall this paper passes the bar for a NeurIPS paper, and we ask the authors to incorporate the promised revisions in their camera ready version.